# Faunal engineering stimulates landscape-scale accretion in southeastern US salt marshes

Sinéad M. Crotty [1,2] ✉, Daniele Pinton [3], Alberto Canestrelli[3], Hallie S. Fischman [1], Collin Ortals[1,3], Nicholas R. Dahl[2], Sydney Williams[1], Tjeerd J. Bouma[4,5] & Christine Angelini[1,3]

The fate of coastal ecosystems depends on their ability to keep pace with sea-level rise—yet projections of accretion widely ignore effects of engineering fauna. Here, we quantify effects of the mussel, *Geukensia demissa*, on southeastern US saltmarsh accretion. Multi-season and -tidal stage surveys, in combination with field experiments, reveal that deposition is 2.8-10.7-times greater on mussel aggregations than any other marsh location. Our Delft-3D-BIVALVES model further predicts that mussels drive substantial changes to both the magnitude ($\pm<0.1$ cm·yr$^{-1}$) and spatial patterning of accretion at marsh domain scales. We explore the validity of model predictions with a multi-year creekshed mussel manipulation of >200,000 mussels and find that this faunal engineer drives far greater changes to relative marsh accretion rates than predicted ($\pm>0.4$ cm·yr$^{-1}$). Thus, we highlight an urgent need for empirical, experimental, and modeling work to resolve the importance of faunal engineers in directly and indirectly modifying the persistence of coastal ecosystems globally.

As rates of sea-level rise accelerate globally, the fate of vegetated coastal ecosystems, such as salt marshes, mangroves, and seagrasses is uncertain[1–6]. Since these systems often occupy a narrow elevational range, relatively small changes in sea level may lead to substantial 'drowning', or conversion of vegetated, intertidal habitat to open water[7–9]. Given the valuable shoreline stabilization, wave attenuation, nutrient filtration, habitat provisioning, and carbon sequestration services provided by these ecosystems[10,11], significant effort and resources have been invested in quantifying their past and current rates of vertical and horizontal accretion and erosion, and using both field data and modeling to forecast their size, stability, and spatial distribution under different sea-level rise scenarios[3,6,12–14]. Historically, modeling efforts primarily focused on the relationships between physical factors including sediment supply, elevation, and tidal range[15]. More recent models also account for the effects of vegetation, such as the trapping of sediment by aboveground stems and leaves, the accumulation of organic matter via root and rhizome production belowground[16], and the feedbacks between vegetation and physical forcing factors[17], in influencing vegetated coastal ecosystem dynamics and stability. Despite the flourishing sophistication of such models, they do not yet consider the role of fauna in altering vertical and horizontal accretion and erosion processes[18,19]. This lack of consideration of faunal influence is potentially problematic given the significant body of literature demonstrating that animals can modify sedimentation processes and vegetation coverage, density, and other above- and belowground traits (e.g., [20]).

[1]Department of Environmental Engineering Sciences, Engineering School for Sustainable Infrastructure and Environment, University of Florida, PO Box 116580, Gainesville, FL 32611, USA. [2]Carbon Containment Lab, School of the Environment, Yale University, 83 Audubon St., New Haven, CT 06510, USA. [3]Department of Civil and Coastal Engineering, School for Sustainable Infrastructure and Environment, University of Florida, PO Box 116580, Gainesville, FL 32611, USA. [4]Department of Estuarine and Delta Systems, Royal Netherlands Institute of Sea Research (NIOZ) and Utrecht University, 4401 NT Yerseke, The Netherlands. [5]Department of Physical Geography, Utrecht University, P.O. Box 80.115, 3508 TC Utrecht, The Netherlands. ✉e-mail: sinead.crotty@yale.edu

Formed in temperate, low energy coastlines around the world[21], salt marshes are among the most well-studied vegetated coastal ecosystems. These intertidal grasslands respond dynamically to tidal oscillations and sea level changes, as periodic flooding and draining of tidewaters control sediment delivery to marsh platforms, as well as plant zonation, productivity, and allocation to above- and below-ground tissues[17]. Zero-, one-, and two-dimensional models have been developed to assess marsh accretion and erosion processes under various environmental scenarios at a single point, along a transect, or across a marsh platform, respectively[17,19,22–26]. More sophisticated ecogeomorphic models of marsh accretion have also been developed to incorporate the direct and indirect feedbacks between salt marsh vegetation and physical processes ([17–19,22–27]; Fig. S1). In these ecogeomorphic models, progressive flocculation and settling of suspended particles as flooding tides breach marsh platforms drives sediment deposition on marsh edges − a process that leads to the formation of levees, or elevated bands of sediment, along the margins of marsh channels and tidal creeks. As tidewaters flood from channels and creeks onto marsh platforms, flow velocities decrease and suspended sediments settle out of the water column, resulting in a sedimentation gradient whereby sediment deposition is highest along marsh margins and decreases with distance onto marsh platforms[28]. This mechanistic understanding of spatial gradients in sediment deposition that contribute to salt marsh vertical accretion is widely supported by empirical data and numerical modeling (e.g., [22–34]).

While salt marshes and other vegetated coastal systems are built and structurally defined by habitat-forming plants, ecosystem engineering infauna and epifauna often play powerful roles in modulating their structure and stability[35–40]. Ecosystem engineering fauna, hereafter 'faunal engineers', can modify sediment deposition and accretion processes through a variety of direct and indirect mechanisms (e.g., [41]). Suspension- and filter-feeding organisms, such as bivalves and sponges, actively contribute to sediment deposition, while bioturbating organisms, such as burrowing crabs and worms, resuspend sediment into the water column[41,42]. Both activity types can directly alter rates of inorganic and organic sediment import and export from coastal systems ([36,43,44] see[45] for biota-mediated blue carbon cycling). Simultaneously, through deposition of nutrient-rich material, oxygenation of soil, and commensurate enhancement of above- and below-ground marsh plant growth, faunal engineers have the potential to indirectly enhance sediment capture by plant leaves or soil accumulation via root production. Further, herbivorous crabs, fungal-farming snails, and other consumers can decrease above and belowground plant biomass directly through their grazing activities, and/or indirectly through the spread of disease[46–48]. Likewise, larger-bodied herbivores (e.g., hogs, sheep, cows, and horses) often trample, graze, and compact marsh vegetation and landforms, thereby altering the spatial patterns in sediment accumulation and accretion or subsidence[49–51]. Most likely because such fauna exhibit significant temporal and spatial variability in their distribution and engineering impacts and are generally perceived to be far less important than plants in controlling sediment deposition at landscape scales, ecogeomorphic models have thus far failed to incorporate their effects.

Across US Atlantic salt marshes, one of the most abundant faunal engineers is the Atlantic ribbed mussel (*Geukensia demissa*, hereafter mussels). Mussels directly increase marsh accretion through their filter-feeding and biodeposition of digested and undigested particles (feces and pseudofeces, respectively), forming localized 'mounds' of sediment that are then slowly conveyed and redistributed across marsh platforms over subsequent tides[44,52]. These nutrient-rich biodeposits locally stimulate the above- and belowground growth of smooth cordgrass (*Spartina alterniflora*, hereafter cordgrass)[37,53]. In mesotidal salt marshes of the southeastern US, mussels aggregate in clusters of up to 200 individuals on marsh platforms, with individual aggregations commonly exceeding 1 m². Previous work[14] elucidated a hierarchy of factors controlling mussel cover at the creekshed (i.e., creek length and associated tidal prism), landscape (i.e., elevation and the associated submergence time), and patch scale (i.e., predation regimes; see Fig. S2). Findings revealed that these biogenic features predictably occur in the highest densities in close proximity to creekheads—where tidewater floods and ebbs from the marsh with each tide—and decrease in density with increasing distance from creekheads onto adjacent marsh platforms[14,44,52]. Therefore, while mussels exhibit patchy spatial cover across landscapes, their presence and densities can be reliably predicted by easily-identifiable tidal creek features[14]. Despite relatively low cumulative coverage at the landscape scale, mussels exert disproportionately strong control over ecosystem functions as they generate local hotspots of cordgrass productivity, macro-invertebrate diversity and biomass, nutrient availability, and infiltration[37,54].

Given that mussels directly and indirectly alter plant growth, sediment deposition, and hydrodynamics, i.e., three key factors in wetland ecogeomorphic models ([17], Fig. S1), and exhibit predictable spatial patterning ([14], Fig. S2), they offer a tractable system for exploring the relative importance of faunal engineers in mediating sediment deposition and accretion processes. Using short-term measurements of deposition across marsh landscapes, tidal phases, and seasons (Fig. 1, panel 1), in combination with manipulative field experiments replicated across marsh elevations (Fig. 1, panels 2–3), we quantify the spatially- and temporally-explicit contribution of mussel populations to sediment deposition. We then define and parameterize a Delft3D-BIVALVES spatial model with our empirical field measurements to disentangle the relative importance of vegetation (cordgrass) and mussels in controlling marsh sediment deposition and accretion at local (1 m²), creekhead (2500 m²), and landscape scales (>10,000 m²). Finally, we explore the validity of the spatial model in predicting real-world effects of mussels on landscape-scale marsh accretion by establishing and monitoring 10,000 m² experimental mussel removal, mussel addition, and unmanipulated control plots for three years (Fig. 1, panel 4). Our findings highlight the powerful local- and landscape-scale influence of mussels in modifying sediment deposition and, ultimately, the vertical accretion of salt marshes in this region. Thus, this work both sets the foundation and highlights an urgent need for further empirical, experimental, and modeling work dedicated to resolving the importance of faunal engineers in modifying vegetated coastal system persistence in this region and across coastlines globally.

## Results
### Regional context
The South Atlantic Bight (SAB) region, extending from Cape Fear (NC) to Cape Canaveral (FL), contains the most expansive salt marsh systems in the U.S.−totaling approximately 750,000 acres. The outer margins of the SAB are microtidal, but they quickly become mesotidal approaching the head of the Bight[55]. River discharge contributes a substantial and continuous supply of suspended fine grain sediment to support vertical accretion of these salt marshes[56]. To first characterize tidal creekhead density across the SAB, we quantified the total number of tidal creekheads in 1 km² marsh areas at 10 sites spanning >200 miles of coastline from Cape Romain, South Carolina to Amelia Island, Florida. Creekhead density was 110.2 ± 6.3 creeks km⁻² (mean ± SD, here and below) and there were no significant differences between the northern (106.5 ± 19.7 creeks km⁻²), Sapelo Island (study site; 98.3 ± 13.1 creeks km⁻²), and southern sites (127.0 ± 13.9 creeks km⁻²; Fig. 2a, b; $p > 0.05$). Given that the creekhead mussel 'halo' extends approximately 50 m into the marsh[14], we define creekhead area of influence to be a 50 m x 50 m area oriented perpendicular to the creek and with the midpoint of one side at the main point of creek entry onto the marsh. Based on this estimate, we calculate creekhead area to be 27.6 ± 5.0% of the total marsh area in this region (Fig. 2c, light gray bars).

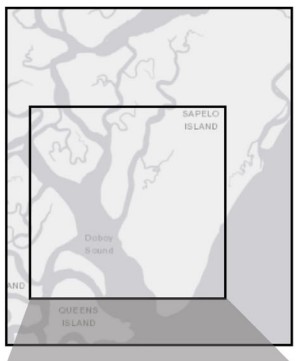

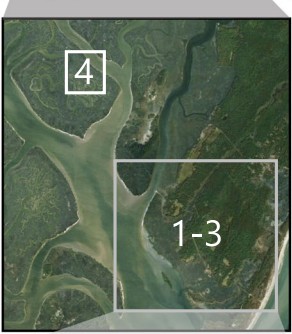

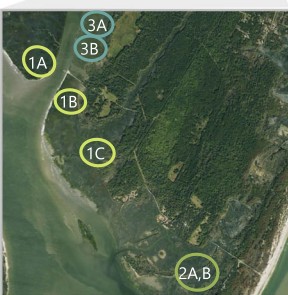

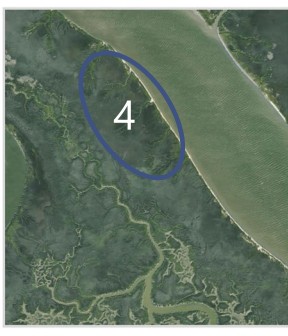

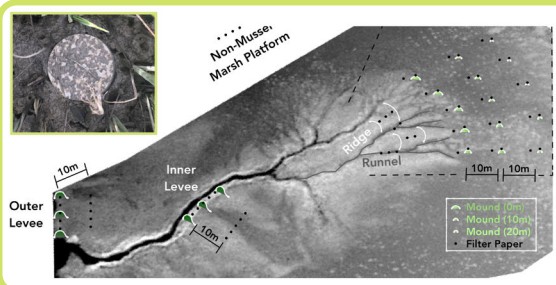

**1a–c.** Landscape Assays of Deposition

**Description:** Measured local-scale sediment deposition using 9-cm diameter filters deployed across 13 marsh area types in 4 × 24-hour deployments (summer spring, summer neap, winter spring, & winter neap)

**Hypothesis:** Local sediment deposition (cm⁻²) is as high on mussel aggregations as other landscape maxima (e.g., outer marsh levees)

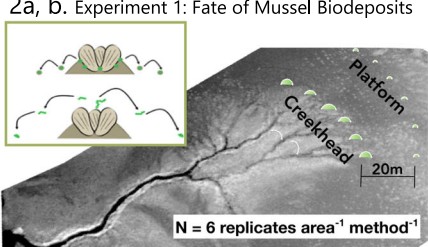

**2a, b.** Experiment 1: Fate of Mussel Biodeposits

**Description:** Tracked fate of A) previously-settled and B) newly ejected, fluorescently-tagged biodeposits on mussel aggregations in each of two marsh areas types (creekhead & platform; n = 6 area⁻¹) over 24-hour deployment

**Hypothesis:** Biodeposits are rapidly redistributed across marsh platforms in all directions (cm-m day⁻¹)

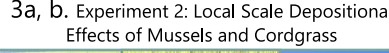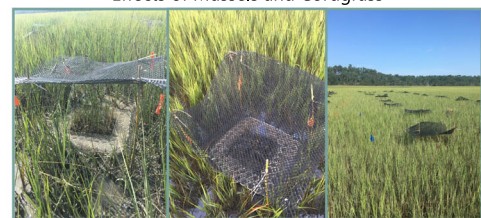

**3a, b.** Experiment 2: Local Scale Depositional Effects of Mussels and Cordgrass

**Description:** Deployed sediment catchment bins (61cm × 61cm) around natural mussel aggregations (36cm × 36cm; 1, 20, 50, and 80 mussels) with and without cordgrass for one month in 2 marsh area types (creekhead & platform)

**Hypothesis:** At intermediate temporal scales (1-month), and across marsh area types, mussel biomass will drive sediment deposition at the patch (m²) scale

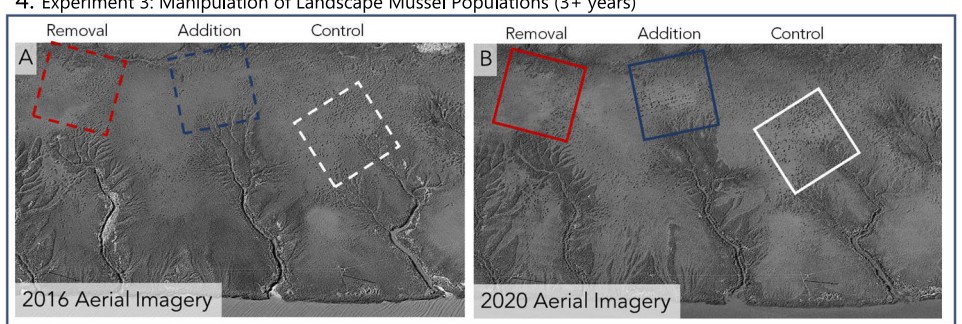

**4.** Experiment 3: Manipulation of Landscape Mussel Populations (3+ years)

**Description:** Removed all mussels from the 2,500m² creekhead (area denoted by dotted lines) of the 'Removal' creek, transplanted them to the 'Addition' creek, and monitored changes to elevation over 3 years

**Hypothesis:** Over multiple years, mussel addition enhances marsh platform accretion relative to the unmanipulated control creek, while removing mussels inhibits accretion**
** We note that additional hypotheses will be generated upon completion of the Delft3D-Bivalve model

**Fig. 1 | Conceptual figure outlining the spatial and temporal scale of all field components and associated hypotheses. 1a–c** Landscape assays of sediment deposition (i.e., 9 cm filter papers;[57]) were distributed across 13 area types over four 24 h tidal deployments. We hypothesized that sediment deposition atop mussel aggregations would be as high as deposition on levee crests. (**2a, b**) Experiment 1 involved tracking the fate of fluorescently tagged previously-settled and newly ejected biodeposits from mussel aggregations (24 h deployments). We hypothesized that sediment would be rapidly redistributed across marsh platforms from these local hot spots of deposition. **3a, b** Experiment 2 involved the deployment of seven treatments containing a range of mussel and cordgrass biomass. Treatments were deployed at the creekhead and on the marsh platform in sediment catchment devices, designed to capture all sediment deposited throughout the 1-month deployment. We hypothesized that mussel biomass would drive sediment deposition at this intermediate temporal and spatial scale. (**4**) Experiment three involved the removal of mussels from one tidal creekhead and the transplantation of these mussels to another proximate creekhead. We hypothesized that the removal of mussels inhibits accretion at the landscape scale, while addition increases it relative to an unmanipulated control. Locations of each experiment are highlighted in the panels at left. Numbers and colors correspond to the experiment of relevance.

To next characterize mussel coverage across the SAB, we surveyed 12 total sites spanning >150 miles of coastline from Edisto Beach, South Carolina to Amelia Island, Florida. Throughout the SAB, mussel aggregations comprise 10.3 ± 2.1% (mean ± SD, here and below) of total creekhead area (Fig. 2a, b). There were no significant differences in mussel coverage of tidal creekheads across the northern (9.3 ± 0.8%), Sapelo Island (11.3 ± 1.0%), and southern sites (9.6 ± 3.2%; $p > 0.05$). When scaled to the marsh creekshed (i.e., not simply tidal creekheads), we calculate mussel areal coverage to be 2.8 ± 0.6% of the total marsh area in the region (Fig. 2c). Based on these results, we suggest that our study site, Sapelo Island, is a salt marsh system representative of the broader SAB region with regards to creekhead density, creekhead area, and mussel cover at the landscape scale.

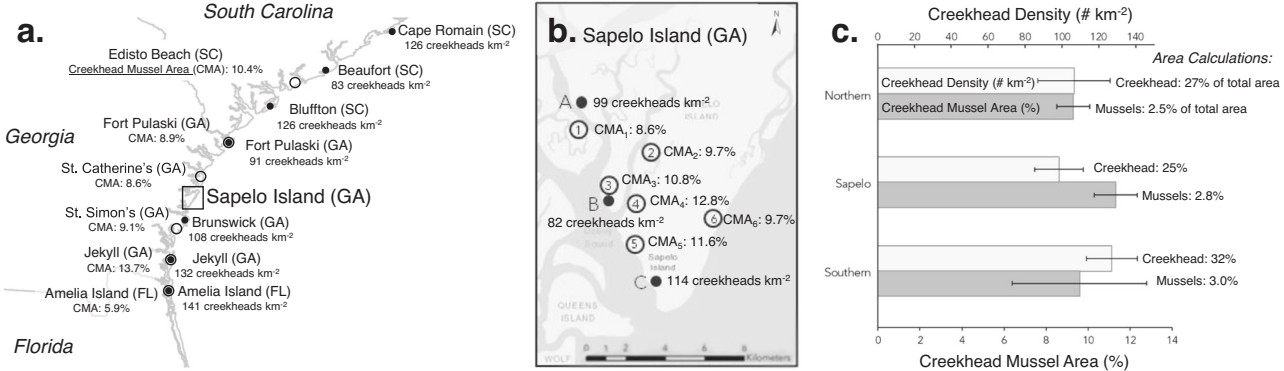

**Fig. 2 | Creekhead density and mussel areal coverage at sites distributed across the South Atlantic Bight. a** Regional surveys quantified the percent area of tidal creekheads occupied by mussels (values on the left of Panel **a** with sites denoted in open circles; $n = 6$ sites) and the density of tidal creekheads within larger creekshed areas (# 1 km$^{-2}$; values on the right of Panel A with sites denoted in black circles; $n = 7$ sites). **b** Sapelo Island sites were similarly assessed for creekhead mussel area (open circles; $n = 6$ sites) and creekhead density (black circles; $n = 3$ sites). **c** There were no statistically significant differences in creekhead density (light grey bars) or creekhead mussel area (dark gray bars) across northern sites ($n_{Creekhead\ Density} = 4$; $n_{Creekhead\ Mussel\ Area} = 3$), Sapelo Island sites ($n_{Creekhead\ Density} = 3$; $n_{Creekhead\ Mussel\ Area} = 6$), and southern sites ($n_{Creekhead\ Density} = 3$; $n_{Creekhead\ Mussel\ Area} = 3$; mean + SE). Finally, assuming a creekhead area of 2500 m$^2$, areal coverage of creekheads and of mussels range from 25–32% and 2.5–3.0%, respectively across the region (mean values reported to right of error bars).

## Landscape assays of sediment deposition over seasons and tidal phases

To quantify and contextualize the depositional effects of mussel aggregations relative to other high-depositional environments (i.e., outer marsh leveed and inner creek levees), we measured sedimentation on the marsh surface using filter paper assays (9 cm diameter; [57]) deployed across 13 location types over each of four 24 h periods: Summer Spring tide (August 2017, Max tidal height: +2.5 m), Summer Neap tide (August 2017, +2.1 m), Winter Spring tide (February 2018, +2.5 m), and Winter Neap tide (February 2018, +2.0 m). Filter deployment locations included outer marsh levee and inner creek levee crests, where models of marsh accretion predict maximum rates of deposition (e.g., [17,32,33,58,59]), and adjacent marsh platform areas, as well as on mussel aggregations at increasing distances landward of tidal creekheads, and marsh platform areas adjacent to these locations. We additionally deployed filters at non-mussel high marsh platforms, >50 m from tidal creek entry points or creek margin levees (see Fig. 1, panel 1 for all locations). The mass of sediment deposited over a 24 h period was 2.8- to 10.7-times higher on mussel aggregations than on levee crest filters during summer neap, summer spring, and winter spring tides, (Tukey HSD, $p < 0.001$), with no significant differences across marsh locations observed in winter neap tides ($p > 0.05$, Season*Tide*Location: $F_{12, 825} = 4.7$; $p < 0.0001$, Fig. 3a–d; see Fig. S3 for analysis of organic and inorganic sediment contribution across locations). We note that, when converted to accretion rates[60], the filter paper results range from +0.01 cm yr$^{-1}$ to +0.74 cm yr$^{-1}$, values comparable to those found in the region (−0.4 to +1.0 cm yr$^{-1}$) from surface elevation tables (SET), feldspar marker horizons, 137 Cs, and 210 Pb data points ([61]; see Supplementary Note 1 for calculations).

## Field experiment 1: fate of mussel biodeposits

To test our hypothesis that sediment deposited by mussels is redistributed and may thus supply sediment to marshes beyond the footprint of mussel aggregations, we measured the transport of previously settled, fluorescently-tagged biodeposits (i.e., mix of feces and pseudofeces) sourced from and then redeployed atop mussel aggregations ($n = 6$ replicates in both 'creekhead' zone and the 'marsh platform' zone; Fig. 1). After one spring tidal cycle (+2.3 m), we observed that this fluorescently-tagged material was redistributed at a mean distance of $30.6 \pm 3.4$ cm (mean ± SE) in all directions in both zones from mussel aggregations (zone; $p > 0.50$; Fig. 4a). We also fed individual mussels fluorescently-tagged tidewater, outplanted the mussels, and tracked their newly-ejected, tagged biodeposits over a spring tidal cycle ($n = 6$

replicates zone$^{-1}$). Tagged biodeposits spread an average of $17.5 \pm 3.3$ cm (Fig. 4b) in all directions from their mussel source, with no significant differences in redistribution distance between creekhead and marsh platform zones ($p > 0.50$). Together, these findings reveal that, over the single-tide timescale of these experiments, mussel biodeposits are: 1) redistributed rapidly; and 2) are at least partially retained within the marsh (and not exported completely with the receding tide). Due to challenges in confidently detecting the spatial extent of the fluorescently-tagged material beyond the single-tide duration of these experiments, additional work is needed to resolve how mussel biodeposits may be redistributed across the marsh over longer time scales.

## Field experiment 2: local scale depositional effects of mussels and cordgrass

To quantify the separate and combined effects of cordgrass and mussels on local sediment deposition across marsh elevations over an intermediate timescale, we deployed transplants of cordgrass and mussel aggregations in sediment catchment devices. Catchment devices consisted of two units, a central unit containing the marsh block with mussels and/or cordgrass, and an outer unit (61 cm x 61 cm) sized to capture biodeposits and/or sediment within vertically-oriented and tightly aggregated PVC traps, and prevent sediment resuspension (Fig. S4). There were seven treatments ($n = 5$ replicates per treatment per marsh zone) in which we varied mussel (M) presence and density, as well as cordgrass (C) presence. The full set of seven treatments included: 1) no-mussel, no-cordgrass controls (0 M, 0 C); 2) cordgrass-only controls (0 M, C +); 3) 1-mussel (1 M, 0 C) blocks; 4) small mussel aggregations (20 M, 0 C); 5) intermediate size mussel aggregations (50 M, 0 C); 6) intermediate size mussel aggregations plus cordgrass (50 M, C +); and 7) large mussel aggregations (80 M, 0 C; Fig. S5). Treatments were replicated in two zones at Airport Marsh on Sapelo Island, Georgia, USA, the creekhead (31°25'28.1"N 81°17'30.2"W) and the marsh platform(31°25'25.3"N 81°17'29.8"W; 70 plots overall). The experiment ran from July 18 to August 18, 2017.

Multiple regression analyses (Creekhead: $F_{2, 32} = 56.2$; $p < 0.0001$; Adj. $R^2 = 0.76$; Platform: $F_{2, 32} = 53.2$; $p < 0.0001$; Adj. $R^2 = 0.75$; see Table S1 for full model results) revealed that mussel biomass is strongly and positively correlated with sediment deposition in both marsh location types (Fig. 4c). After standardizing sediment deposition by mussel biomass (in g sediment deposited per g mussel), we found that the effect of mussels was 2-times greater at the creekhead than marsh platform, reflecting gradients across marsh locations in tidal

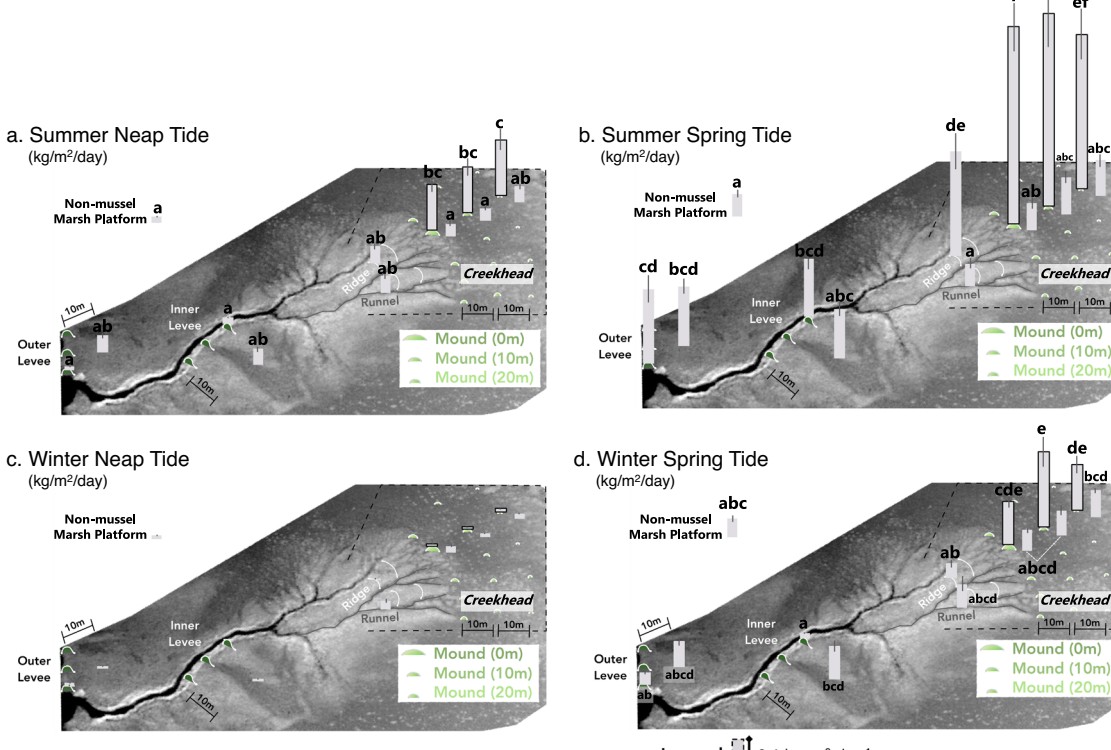

**Fig. 3 | Filter paper results. a** Summer neap, **b** summer spring, **c** winter neap, and **d** winter spring tide results are presented across 13 marsh location types ($n = 15$ filters/location/tide). Mean sediment deposition is presented as gray bars (mean ± SE) on each marsh location, with letters denoting statistically significant differences among treatments (Season*Tide*Location: $F_{12, 825} = 4.7$; $p < 0.0001$). For ease of interpretation, Tukey HSD post hoc analyses are conducted separately for each season using a corrected $p$-value of $p < 0.001$. Results collected from atop mussel aggregations are presented as gray bars with a black border (at 0 m, 10 m, and 20 m from the tidal creekhead). Results from locations not associated with mussel aggregations (i.e., all other marsh location types) are presented as gray bars with no border.

submergence, the period over which mussels suspension-feed and produce biodeposits. In contrast, local cordgrass removal had no effect on sediment deposition in either zone in the two experimental treatments replicated with and without cordgrass (Creekhead: $p = 0.29$; Platform: $p = 0.99$, see dashed line in Fig. 4c). These experimental results re-enforce the landscape filter assays in highlighting that mussels, through their active filtration of tidewater and production of biodeposits, drive significantly more sediment deposition than cordgrass at the scale of our experimental plots and that these effects are amplified at lower elevation tidal creekheads relative to higher elevation marsh platforms. To accurately capture the effects of vegetation presence beyond the patch scale, however, experimental removal of vegetation would need to occur over much larger areas (e.g.,[62]).

**Delft 3D model**
To explore the potential for mussels to modify salt marsh sediment budgets and vertical accretion at temporal and spatial scales infeasible to quantify in landscape assays or field experiments, we performed numerical simulations using the Delft3D-FLOW model[63,64]. Delft3D-FLOW solves the Navier-Stokes equations for an incompressible fluid under the shallow water assumption and the Boussinesq approximation in a boundary-fitted grid. Sediment transport is computed by solving the advection-diffusion equations for suspended sediment. We expanded the source code of the model by adding a new module, termed Delft3D-BIVALVES, to simulate sediment filtration and deposition processes associated with mussels in the case of this study, or other filter-feeding, biodepositing fauna. The model domain defines the salt marsh as a rectangular system, in which the marsh platform is

connected to the main channel by a tidal creek. The domain extends 50 m and 207 m in the long-shore and landward directions, respectively (Fig. 5a). In this model domain, mussel aggregations occupy only the 'creekhead', which is the 50 m by 50 m area between the creek and the upper boundary of the domain. While a simplified version of a salt marsh, the proportionate cover of tidal creek, creekhead area, marsh platform and mussels are generally representative of salt marshes in the region (i.e., creekhead and mussel aggregations occupy 27% and 2.7% of total marsh area, respectively).

In this study, we considered two vegetation scenarios (i.e., with or without cordgrass) fully crossed with three mussel configuration scenarios [i.e., mussel aggregations occupying 0%, 10% (ambient densities), and 20% of the creekhead area] resulting in six scenarios overall (see Supplementary Notes 2-3 for creek and mussel density sensitivity analyses). For each scenario, we estimated annual sediment deposition (m³ yr⁻¹) and vertical accretion (cm yr⁻¹) at five 'local' marsh location types (i.e., 1 m² areas in levee crest, levee-adjacent, mussel aggregation, aggregation-adjacent, and non-mussel platform locations) to complement the landscape assay and field experiment aspects of this study. For the three vegetated scenarios (0, 10 and 20% mussel cover), we also quantified sediment deposition and vertical accretion at the whole creekhead (2500·m²) and at the entire marsh domain (10,350 m²) scale to evaluate landscape-scale effects of mussels. At the three scales, we compare sediment deposition and marsh accretion patterns in each cordgrass mussel scenario relative to a 'baseline' scenario defined by cordgrass present and ambient (10%) mussel densities (i.e., a natural marsh supporting both the foundational plant and faunal engineers) to facilitate interpretation of the model results. Finally, to evaluate the relative importance of the gradual spreading, or conveyance of mussel

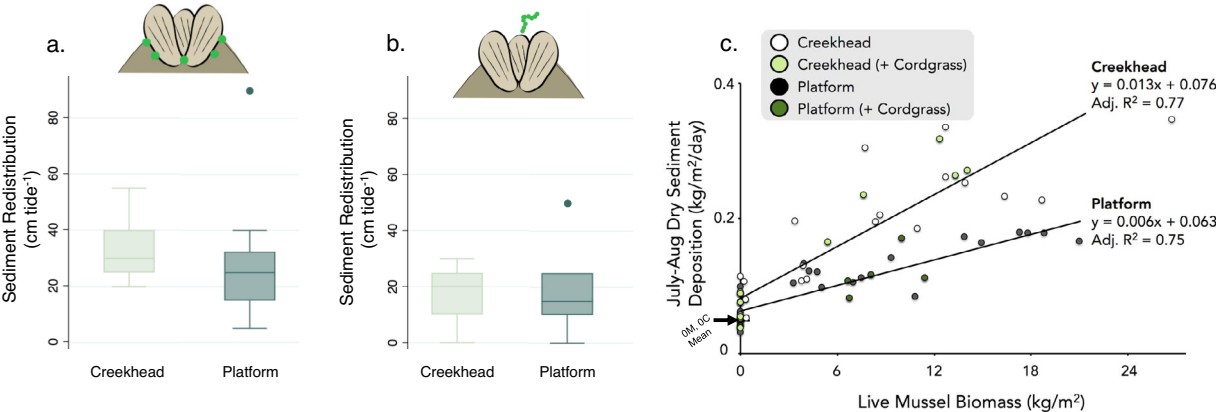

**Fig. 4 | Results from experiments 1 & 2.** Box plots of sediment redistribution (in cm traveled per tide) is shown of previously settled biodeposits (**a**) and newly ejected material (**b**) in both the creekhead (light green) and marsh platform (dark green) zones ($n = 6$ mounds/zone). Box plots present minimum, 25th percentile, median, 75th percentile, and maximum as lines. Outliers are presented as solid circles. **c** The relationship between live mussel biomass (x axis) and sediment biodeposition (y axis) is presented in both the creekhead and on the marsh platform. Model summaries are presented inset with adjusted $R^2$ values. For reference to no-mussel, no-cordgrass control treatments (0 M, 0 C), we present the mean value of this treatment on the y axis. Month-long dry sediment deposition is presented in units of kg/m²/day.

biodeposits, in affecting the strength of faunal engineering effects on landscape-scale sediment deposition, we ran a sensitivity analysis in which we allow for the resuspension of material deposited on mussel aggregations. All model simulations encompassed 16-day long periods, representing a yearly average Spring+Neap cycle[65]. Annual values are calculated as the product of the results obtained from the 16-day simulations and the number of cycles present in a year.

In the baseline scenario, the highest sediment accretion at the local scale (1 m²) occurs on mussel aggregations ($3.1 \pm 1.4$ cm yr⁻¹; mean ± SD) relative to all other location types ($0.9 \pm 0.2$ cm yr⁻¹; Fig. 5b). Compared to the baseline, the removal of cordgrass throughout the marsh domain elicits an increase in sediment deposition and vertical accretion in levee crest ($59.7 \pm 0.4\%$ increase), levee-adjacent ($34.2 \pm 2.3\%$ increase), mussel aggregation ($14.0 \pm 2.0\%$ increase), mussel-adjacent ($24.1 \pm 14.0\%$ increase), and marsh platform locations ($14.5 \pm 3.4\%$ increase; Fig. 5c). While the most emphasized bio-geomorphic feedback of vegetation consists of increased friction, reduced velocity, and increased deposition[66–68], this negative effect of vegetation on deposition has been observed both numerically and experimentally[34,69–72]. Mechanistically, the negative effect of vegetation occurs when the aboveground biomass diverts flow and sediments toward less vegetated areas or reduces the exchange of water and sediments between high and low elevation areas of the system. In contrast, the removal of mussels throughout the marsh domain leads to a decrease ($-67 \pm 2.1\%$) in local scale deposition and accretion on mussel aggregation locations in both the presence and absence of vegetation, but has minimal effect on deposition and accretion at other marsh location types (Fig. 5c). At this local scale, doubling mussel densities (20% of tidal creekhead area) paradoxically results in a slight decrease in mean deposition and accretion on mussel aggregations ($-9.2 \pm 7.0\%$) relative to the baseline scenario.

At the creekhead (2500 m²) scale, the removal of mussels (0% cover) decreased total sediment deposition by $-1.92$ m³yr⁻¹ and vertical accretion by $-0.08$ cm yr⁻¹ relative to the baseline 10% mussel scenario. In contrast, the doubling of mussel populations (20%) relative to baseline conditions enhanced creekhead sediment deposition by $+0.81$ m³yr⁻¹ and vertical accretion by $+0.03$ cm yr⁻¹ (Fig. 5d, e). At the landscape scale (10,350 m²), the removal of mussels (0% cover) at the tidal creekhead decreased total sediment deposition by $-0.92$ m³yr⁻¹ and vertical accretion by $-0.01$ cm yr⁻¹ relative to the baseline 10% mussel scenario. In contrast, the doubling of mussel populations (20%) relative to baseline conditions enhanced creekhead sediment

deposition by $+0.79$ m³yr⁻¹ and vertical accretion by $+0.01$ cm yr⁻¹ (Fig. 5d, e). The effects of both mussel removal and addition were concentrated in the creekhead region of the domain. Therefore, while levees are the only locations in the marsh that support particularly high deposition when mussels are absent, we find that the entire creekhead region becomes an accretion and deposition hotspot when mussels are added. Finally, to assess the role of resuspension of sediment deposited on mussel aggregations, we set a critical shear stress for erosion ($\tau_{c,e}$) equal to $1$ Nm⁻² on mussel aggregations[73]. Sensitivity analysis revealed that resuspension increases sediment spread through the marsh (i.e., from atop mussel aggregations to other inland marsh location types such as marsh platforms) as predicted. However, a small percentage (<1%) of resuspended sediment exits the marsh through the creek. In either case, there is a significant impact of mussel mounds on sediment trapping in the marsh platform. These model results suggest that the presence and population size of this faunal engineer can drive predictable spatial patterns in sediment deposition and vertical accretion across marsh landscapes.

## Field experiment 3: manipulation of landscape mussel populations

To explore the validity of the model predictions and assess the potential effects of mussel populations on salt marsh landscape accretion, we removed approximately 200,000 mussels from one tidal creekhead and transplanted them to another creekhead in summer 2017 (full manipulation plot size: 10,000 m²; creekhead plot size: 2500 m²). We additionally established an unmanipulated control creekhead in the same marsh site. Due to logistical and permitting constraints, it was not feasible to replicate the treatments across multiple sites; instead, the three plots occupied a single contiguous creekshed (Fig. 6a, b). Initial marsh platform elevation ($n = 86$ total points) did not differ across our control ($0.798 \pm 0.026$ m AMSL; all mean ± SD), mussel removal ($0.804 \pm 0.045$ m AMSL), and mussel addition ($0.805 \pm 0.075$ m AMSL) creekheads. In addition, since mussel aggregations exhibit a height ceiling of $+0.84$ m AMSL[14], we utilized mussel mound height calculations to provide a second estimate of initial elevation across the creekshed (Fig. S7). Across >60 mounds per creekhead ($n = 250$ total), these results reveal that prior to treatment initiation, the removal creek marsh platform was 1.7 cm lower in elevation ($0.742 \pm 0.030$ m AMSL) than the control creekhead ($0.759 \pm 0.030$ m AMSL; $p < 0.05$), which was similar to the addition creekhead ($0.764 \pm 0.024$ m AMSL; $p > 0.05$).

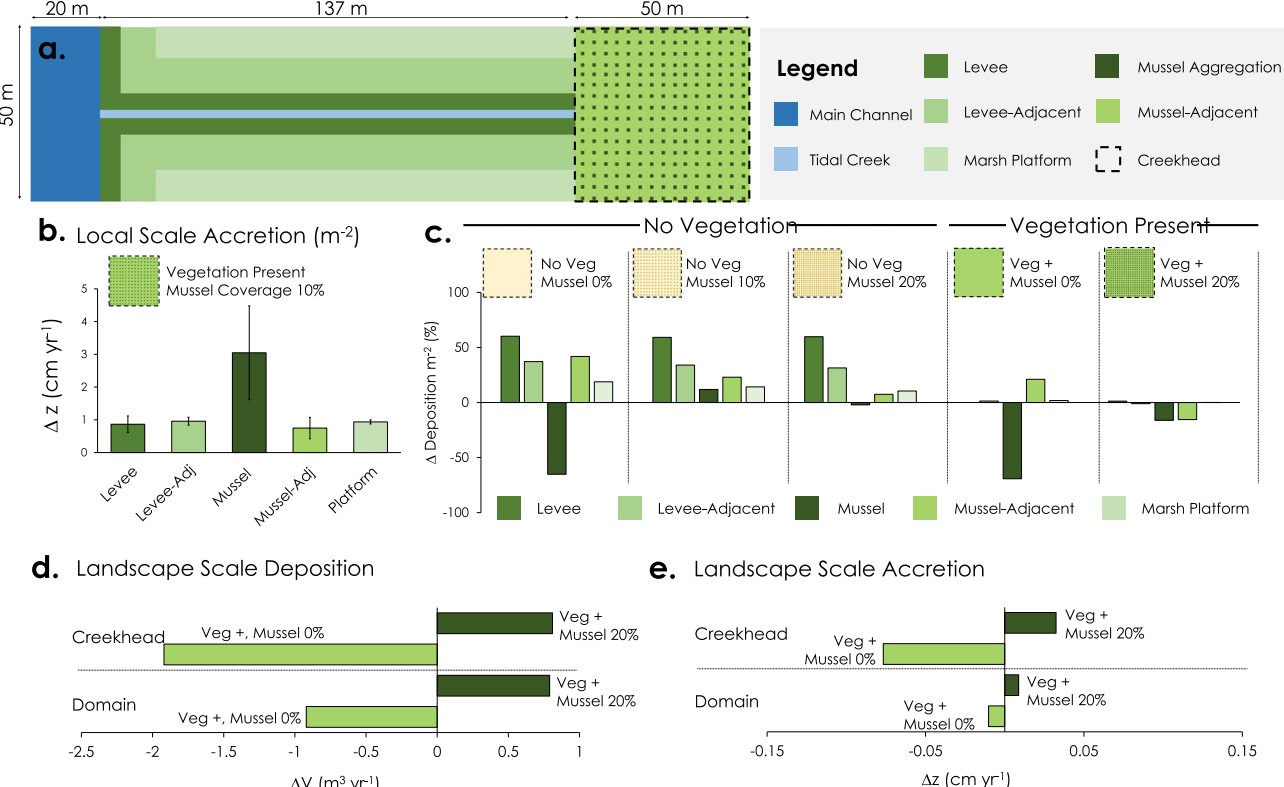

**Fig. 5 | Delft3D-BIVALVES Model Results. a** Model domain and area types are delineated and color coated to depict locations of different vegetation and mussel patterning. **b** Mean local scale accretion (1 m$^2$) of the five marsh area types in the baseline scenario (defined as vegetation present, mussel coverage 10%) are shown (mean ± SD; $n > 150$ per area type, depending on number of cells occupied by area type). For each marsh location type and scenario, we then calculate percent change in local deposition (m$^{-2}$) compared to the baseline scenario in (**c**). Change in landscape scale deposition (**d**) and accretion (**e**) from baseline scenarios are presented for both mussel removal (0%) and mussel addition (20%) scenarios in the creekhead (top panels) as well as in the entire domain (bottom panels).

After three years of monitoring, we developed a digital elevation model (DEM) from drone imagery collected in 2020, and filtered these data to remove vegetation (Fig. 6c). The 2020 DEMs, accounting for initial elevation differences, revealed that relative to the unmanipulated control creekhead (2500 m$^2$), the mussel removal creekhead was 5.0 ± 0.5 cm lower in elevation (−1.7 cm yr$^{-1}$), while the mussel addition creekhead was 1.2 ± 0.8 cm higher in elevation (+0.4 cm yr$^{-1}$; Fig. 6d). Across marsh landscapes, rates of salt marsh vertical accretion range from −0.4 to +1.0 cm yr$^{-1}$ (mean ± SE; 0.2 ± 0.03 cm yr$^{-1}$) in the region[61]. Thus, while we acknowledge that effects of mussels are likely to be concentrated in the area measured (2500 m$^2$ creekhead), and were in part driven by the loss of mussel mound physical structure and the loss of the associated benthic infauna (e.g., bioturbating crabs), changes to rates of vertical accretion associated with the mussel treatments were substantial. Specifically, mussel removal amplified an existing, subtle difference in elevation and caused the creekhead without mussels to lose elevation relative to the adjacent marsh platform area, while mussel addition bolstered marsh surface elevation gains relative to controls. Further, effects of mussel removal and addition are ≥5-fold greater than the predictions of the Delft-3D-BIVALVES model, which solely quantifies direct effects on sediment deposition, and assumes no conveyance of biodeposits across marsh landscapes. These results, therefore, highlight that mussel biodeposition, in combination with their indirect effects on accretion processes (e.g., enhanced primary productivity, sediment stabilization, etc.), drive substantial and measurable changes to marsh geomorphology over several year time scales—and may therefore play a key role in helping marshes keep pace with sea-level rise.

## Discussion

Overall, our results reveal that mussels harvest sediment from the water column through their filtration activities and act as important, spatially distributed epicenters for sediment input into this vegetated coastal ecosystem. We demonstrate that these faunal engineers substantially contribute to salt marsh sediment budgets, locally enhancing deposition by up to an order of magnitude. Our implementation of the Delft-3D-BIVALVES module and subsequent simulations further emphasize that mussels can successfully be incorporated into models of coastal wetland accretion, and together with the large-scale mussel manipulation, we highlight that their landscape effects are demonstrably contributing to marsh vertical growth. Although not explicitly evaluated in this study, we project that mussels, by bolstering sediment deposition and marsh accretion, likely play a key role in modulating the future persistence of these intertidal ecosystems with accelerating sea-level rise. However, further research is needed to address the long-term and large-scale interaction between mussel aggregations, marsh accretion, and sea level –and what the consequences of such interactions are for salt marsh morphodynamics and stability in the face of sea-level rise.

Existing ecogeomorphic models neatly assess the interactive effects of marsh vegetation and geomorphology (Fig. S1; [17,27,33]). The relationship between stem density and shape, the hydrodynamic effects of these features, and the ensuing passive sediment capture and deposition generated by plant tissue are central mechanisms highlighted under this conceptual framework[16,74,75]. Belowground production of organic material is an additional mechanism through which marsh vegetation can alter marsh geomorphology and contribute to building marsh platform elevation[27,76]. Expanding upon this prevailing framework, our results and prior studies (e.g., [61]) indicate

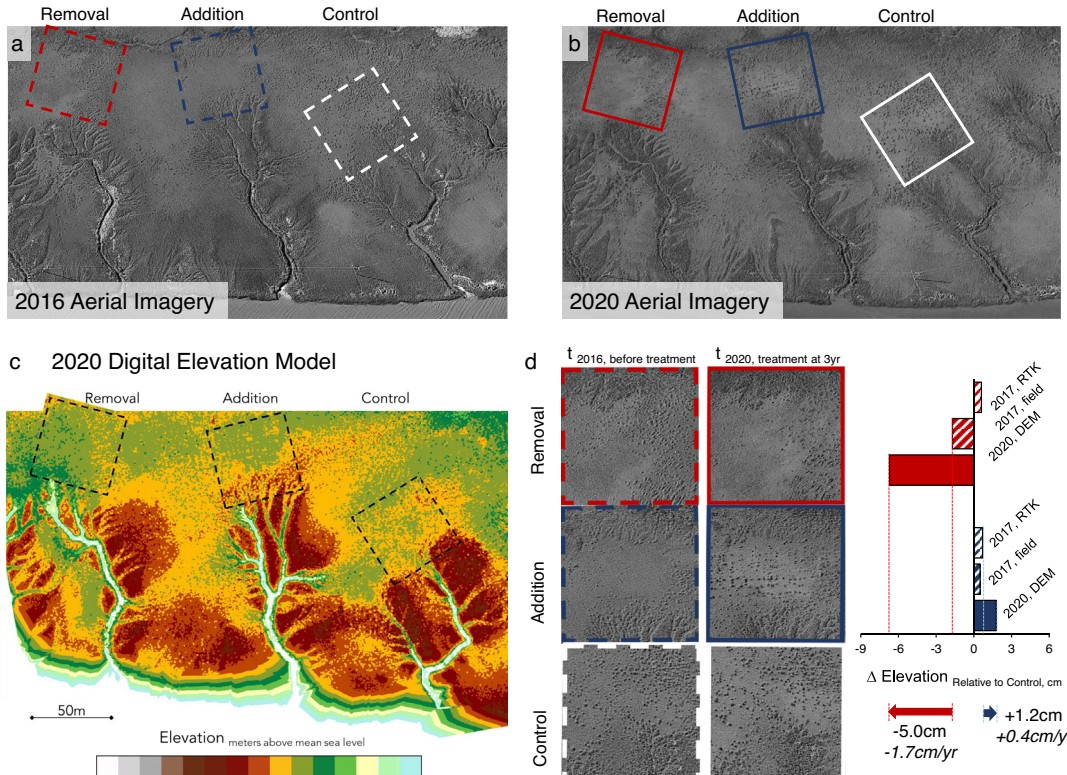

**Fig. 6 | Creekshed Mussel Manipulation.** Prior to deployment in 2017, (**a**) we first delineated the creekhead area (2500 m²) for the mussel removal (highlighted in red, dashed), mussel addition (blue, dashed), and control creeks (white, dashed). Aerial imagery from 2020 (**b**) shows the same experimental areas and visual changes to the condition of the marsh platform (e.g., noticeable grid of mussel aggregations and increased primary productivity on addition creekhead outlined in blue, and loss of those features on the removal creekhead outlined in red). The

2020 DEM (**c**) depicts marsh elevation across the entire creekshed (red colors indicate higher elevations, shifting down to orange, yellow, and green at lower elevations). We then compare differences in elevation between the control and each of the experimental treatment plots from two methods in 2017 and from the DEM creekshed area in 2020 (**d**). Results highlight that, relative to the control trajectory, the mussel removal creek lost elevation (−1.7 cm yr⁻¹) while the mussel addition creek gained elevation (+0.4 cm yr⁻¹).

that engineering fauna importantly alter these mechanisms through both direct and indirect effects (Fig. 7a, b). Mussels, through the deposition of nutrient-rich feces and pseudofeces, indirectly enhance above and belowground vegetation biomass, as well as alter stem density and shape to alter both tidal hydrodynamics (e.g., water flow and turbulence at the local scale) as well as the mechanistic pathways well-described in the literature on marsh accretion (e.g., [37,38,53]). In addition, through their active filtration of tidewaters, binding of small, suspended particles into aggregates, and expulsion of these mucus-bound aggregates that sink and stick to the marsh surface, mussels directly enhance rates of deposition, a mechanism hypothesized to actively build marsh platform elevation[44,77,78]. We note that such indirect effects are likely to be substantial, as the landscape-scale experimental mussel manipulation results were far larger in magnitude than those predicted by models focused on solely direct effects.

Based on the significant contribution of mussel engineering effects to sediment deposition, as well as the ability of their contributions to be predicted over space and time, we argue that integrating this faunal engineer into ecogeomorphic models of southeastern US salt marshes constitutes a necessary and feasible next step for more accurately evaluating their response to present and future rates of sea-level rise. More importantly, the incorporation of spatially-explicit effects of faunal engineers will increase model utility, allowing for predictions to be adapted based on simple geomorphological metrics of coastal landscapes. Local conservation or management efforts can therefore more readily self-assess landscape susceptibility to drowning in relevant locations, results that have the potential to inform and motivate more local-scale conservation efforts.

One tractable pathway for such expansion is to further employ the Delft-3D model and adapt the BIVALVES module we develop in this study to probe questions about the future geomorphic evolution of these marshes and other vegetated coastal ecosystems where faunal engineers influence plant, sediment and hydrodynamic processes. In particular, there may be particular value in exploring the relative importance of bivalves and other faunal engineers in modifying the accretion processes and rates of organogenic coastal wetlands, where land formation and growth primarily depend on plant growth (particularly of roots and rhizomes) and secondarily upon sediment deposition, versus minerogenic wetlands that are more strongly dependent on sediment deposition.

A variety of functional groups of engineers that alter sediment dynamics are likely to exert strong, albeit varied, effects on factors relating to both vegetation ecology and geomorphology across coastal ecosystems. For example, endobenthic invertebrates, such as lugworms (*Arenicola marina*) and burrowing ghost shrimps (*Callianassa* sp.), commonly act as bioturbators, altering sediment cohesion, stability, and resuspension, as well as nutrient chemistry through their continued processing and displacement of sediment (Fig. 7c; [36,79,80]). Through these activities, bioturbators may play a key role in broadly conveying sediment captured by filter and suspension-feeders—although the magnitude and direction of these interactions are likely context dependent. Further, faunal grazers, such as the herbivorous purple marsh crab (*Sesarma reticulatum*, Fig. 7d; [48]) or domestic cattle[81], likely exert strong effects on accretion processes through direct effects on vegetation density, structure, and tissue allocation. These effects may be positive at low grazer densities, and shift to being

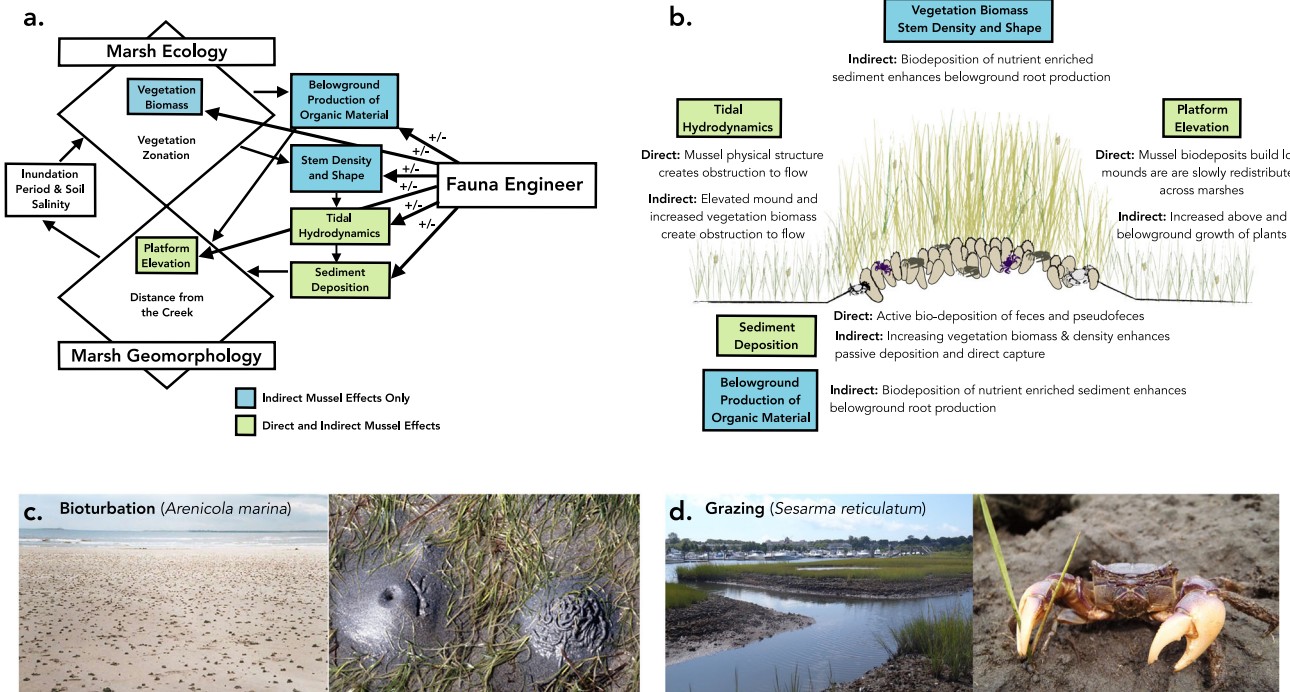

**Fig. 7 | Fauna engineering effects on ecogeomorphology of vegetated coastal ecosystems. a** Conceptual model (adapted after Fagherazzi et al.[27]) depicting the mechanisms through which fauna engineers alter ecogeomorphology, with direct and indirect mussel effects highlighted (blue and green boxes, inset). **b** Mussel engineering effects on marsh ecogeomorphology are illustrated and further described. Other fauna engineers likely alter vegetated coastal ecosystem accretion processes, such as through **c** bioturbating effects of lugworms (*Arenicola marina*) and **d** above and belowground grazing effects of the omnivorous marsh crab, *Sesarma reticulatum*.

strongly negative at higher grazer densities, a phenomenon observed across ecological systems (e.g.,[82]). Such density-dependent effects of grazers highlight a final point: understanding the spatial patterning and density of fauna is necessary to quantify and predict current and future effects of these engineering organisms. In many cases, including with mussels in southeast US salt marshes, spatial patterning is strongly linked to the underlying system geomorphology, as it structures the delivery of food and larvae, as well as spatial and temporal gradients of physical and biological stress[14]. Ultimately, the informed and thoughtful incorporation of faunal engineering effects into eco-geomorphic models and conservation planning efforts will be an important step forward to better predict and enhance the resiliency of coastal ecosystem to sea-level rise.

## Methods
### Regional context
To understand the variation in salt marsh geomorphology and mussel coverage across the South Atlantic Bight (SAB), we assessed density and areal coverage of[1] tidal creekheads and[2] mussel aggregations with a combination of published data and new field surveys across the region. First, to assess creekhead density, we selected 10 sites ranging from Cape Romain (SC) to Amelia Island (FL). Given that all of our experiments were conducted on Sapelo Island and the surrounding marsh islands, we selected three sites on Sapelo Island for comparison with four sites to the north and three to the south[61]. At each site, we scored the total number of tidal creekheads in a 1 km² contiguous marsh area using Google Earth. Assuming each tidal creekhead constitutes approximately 0.0025 km², we calculate the creekhead areal coverage to be:

$$Creekhead\ Areal\ Coverage\ (\%) = \frac{(0.0025 km^2) \times Creekhead\ Density\,(\#km^{-2})}{Marsh\ Creekshed\ Area\,(1km^2)} \times 100\%$$

(1)

Differences across northern, Sapelo Island, and southern sites were assessed with a one-way ANOVA with location as the main factor.

To next test the hypothesis that creekhead mussel coverage is similar at sites across the SAB, we conducted surveys of mussel aggregations at 12 sites across the region from Edisto Beach (SC) to Amelia Island (FL). Previous work[14] has shown that mussel aggregations decrease in size and density with increasing distance from the tidal creekhead, so we focused our measurements at three distances from one tidal creekhead onto the marsh platform: 0 m, 20 m, and 40 m. We note that at all sites, mussel aggregations extended >40 m from the tidal creekhead. Sites were again distributed across the region, and included 3 sites to the north, 3 sites to the south, and 6 sites on Sapelo and its back barrier marsh islands. At each site, we selected one representative creek 100–175 m in length and ensured that the tidal creekhead did not overlap spatially with a tidal creekhead of an adjacent creek. At each distance from creekhead, we established one 50 m x 1 m transect. Walking the transect line, we scored each mussel aggregation, counting the total number of mussels and measuring the mound dimensions (L x W x H). We then calculated the areal coverage of mussels within each transect (50 m²) and took the mean value across the three distances as the measure for the site. All data was collected between May and August in 2016 and 2017. Differences across northern, Sapelo Island, and southern sites were assessed with a one-way ANOVA with location as the main factor. Finally, we calculated creekshed mussel areal coverage in the three sub-regions, as the product of the percent of creekshed occupied by creekheads (sub-region mean, %) and the proportion of creekhead area occupied by mussels at each site.

### Landscape assays of sediment deposition over seasons and tidal phases
To quantify the relative rates of sediment deposition across marsh landscapes, we deployed 9-cm diameter filter papers (Whatman

Quantitative Filter Paper, Grade 42 Circles, Ashless, 90 mm; 57) at 13 location types across 3 sites. Locations included: 1) outer marsh levee ('outer levee'), 2) marsh platform 10 m inland from outer marsh levee ('outer levee-adjacent'), 3) inner tidal creek levee ('inner levee'), 4) marsh platform 10 m inland from inner tidal creek levee ('inner levee-adjacent'), 5) non-mussel marsh platform (>50 m from mussel creek-head), 6,7) ridge/runnel area at tidal creekhead ('ridge' and 'runnel'), 8,9) mussel aggregations and adjacent non-mussel marsh areas at the tidal creekhead ('0 m ON mussel mound' and '0 m OFF mound'), 10,11) mussel aggregations and adjacent marsh areas 10 m onto marsh plat-form from tidal creekhead ('10 m ON mussel mound' and '10 m OFF mound'), and 12,13) mussel aggregations and adjacent marsh areas 20 m onto marsh platform from tidal creekhead ('20 m ON mussel mound' and '20 m OFF mound'). At each location type, we used 15 replicate filters, spaced 1–2 m apart. Each pre-weighed and labeled filter paper was deployed attached to a Polystyrene Petri Dish (100 × 15 mm) using 1.5 mm steel wire. After 24 h in the field, all filters were harvested, dried in an oven at 60 °C, and reweighed. Filter papers were deployed at four tides: Summer Spring (August 2017, +2.5 m), Summer Neap (August 2017, +2.1 m), Winter Spring (February 2018, +2.5 m), and Winter Neap (February 2018, +2.0 m).

To quantify the total and percent inorganic and organic material that was deposited on the marsh surface over a 24 h period, we deployed 8 replicate 4.7 cm diameter filter papers (Whatman Glass Microfiber Filter Paper, Grade GF/F Circles, 47 mm) across five marsh locations at one site. Locations included: 1) outer marsh levee ('outer levee'), 2) marsh platform 10 m inland from outer marsh levee ('outer levee-adjacent'), 3–4) mussel aggregations and adjacent non-mussel marsh areas at the tidal creekhead ('0 m ON mussel mound' and '0 m OFF mound'), and 5) non-mussel marsh platform. Prior to deployment, filter papers were combusted in a 450 °C furnace for 4 h and stored in aluminum foil packets. Packeted filter papers were then labeled and pre-weighed. Once in the field, filter papers were removed from their packet with forceps, placed on a petri dish inserted into the marsh sediment during a Summer Spring low tide (+2.5 m), and secured with 1.5 mm steel wire.

After 24 h, the filter papers were collected with forceps and inserted back in their corresponding packet. Upon transport back to the lab, the packeted filter papers were dried in a 60 °C oven until constant mass was obtained and re-weighed. The change in weight between pre- and post-deployment was used to calculate total dry weight. Packeted filter papers were combusted again in a 450 °C fur-nace for 4 h and re-weighed. The total dry weight and the weight lost from the second combustion were then used to calculate total inor-ganic and organic dry weight and percent organic material for each filter paper.

To calculate the organic and inorganic material in persistent in marsh sediment layers, 5-cm cores were collected from the sediment layer using a 60 mL syringe with a 2.5 cm diameter. Cores were taken at same five location types: levee crest, levee-adjacent, on-mound, mound-adjacent, and non-mussel marsh platforms. Eight cores, 1–2 m apart, were collected from each location and placed into pre-weighed foil packets. Cores were dried at 60 °C in an oven until constant mass was obtained, weighed, and combusted in a 450 °C furnace for 4 h. The cores were then reweighed, and the weight loss after combustion was used to calculate the percent organic (and inorganic) material.

The mass of both organic and inorganic material deposited on each mussel aggregation filter was far greater (0.11 g and 0.50 g, organic and inorganic sediment, respectively, here and below) than that deposited on levee crests (0.02 g and 0.06 g), levee-adjacent (0.04 g and 0.15 g), and non-mussel marsh platforms (0.04 g and 0.19 g; $F_{4,38} = 9.5$; $p < 0.0001$; Adj. $R^2 = 0.47$). The percent organic content of this deposited material was higher on levee crests (22.8%) than on mussel aggregations (18.7%) and non-mussel marsh platforms (18.8%; $F_{4,38} = 9.0$; $p < 0.0001$; Adj. $R^2 = 0.46$; Fig. S3), with intermediate

values in both levee-adjacent (19.9%) and aggregation-adjacent areas (20.1%; $p > 0.05$). Despite these differences in the composition of material deposited on filters, the collection and processing of cores spanning the top 5 cm of marsh sediment across the same five zones revealed that the organic content of this upper layer of the marsh was similar across the landscape ($p > 0.20$), with all locations exhibiting 13–14% organic content (Fig. S3).

### Field experiment 1: fate of mussel biodeposits

To assess the distribution of sediment supplemented by mussels via local biodeposition and, in turn, their contribution to sediment supply across the broader marsh landscape, we measured the transport of previously settled biodeposits as well as those actively deposited over one tidal cycle. For each process, we selected 6 mussel mounds in two marsh zones where mussels commonly aggregate: 1) the creekhead and 2) 20 meters away from the creekhead on the marsh platform. All focal mounds were at least 5 meters apart to avoid mixing of biode-posits. We addressed the transport of previously settled biodeposits by first removing 2 cm of each mound's biodeposit layer, homo-genizing it with fluorescent chalk (Irwin Straight-Line Fluorescent Orange Marking Chalk) at a 2:1 ratio (biodepost:chalk), and evenly distributing the mixture back on the mounds. We then revisited the mounds at night after one tide had flooded over the mounds (max tidal height +2.2 m) and traced the distribution of fluorescent material through black light detection. We measured the maximum distance fluorescent material traveled in each direction to quantify transport of previously settled biodeposits across the marsh landscape.

To account for the distribution of biodeposits ejected by actively filter-feeding mussels, we collected 10 mussels from each mound, transported them back to University of Georgia Marine Institute's wet lab, depurated them in saltwater (Instant Ocean, 28 ppt) for 24 h, and allowed them to feed on a mixture of seawater and fluorescent chalk for 2 h. We then rinsed the mussels to remove any loose fluorescent material from their shells before transplanting them back into the focal mounds at low tide. We then revisited the mounds at night after one tide had flooded over the mounds and traced the distribution of fluorescent material through black light detection. We measured the maximum distance fluorescent material traveled in each direction to quantify transport of actively ejected biodeposits across the marsh landscape.

### Field experiment 2: local scale depositional effects of mussels and cordgrass

The second experimental study was conducted at Airport Marsh on Sapelo Island, Georgia, USA. At this site, the experiment was deployed at two zones: the marsh platform >85 m from the nearest tidal creek (31°25'25.3"N 81°17'29.8"W) and the creekhead, where the tidal creek enters onto the marsh platform and tidal water first floods the marsh (31°25'28.1"N 81°17'30.2"W). Within each zone, we deployed seven experimental treatments ($n = 5$ replicates per treatment per zone) in which we varied mussel (M) presence and density, as well as cordgrass (C) presence. The full set of seven treatments included: 1) no-mussel, no-cordgrass controls (0 M, 0 C); 2) cordgrass-only controls (0 M, C +); 3) 1-mussel (1 M, 0 C) blocks; 4) small mussel aggregations (20 M, 0 C); 5) intermediate size mussel aggregations (50 M, 0 C); 6) inter-mediate size mussel aggregations plus cordgrass (50 M, C +); and 7) large mussel aggregations (80 M, 0 C; Fig. S5).

In July 2017, we harvested 70 blocks of marsh peat (50 cm x 50 cm x 20 cm) from the experimental site using flat-edge shovels. We selected 30 blocks of standardized cordgrass density (48.9 ± 9.0 g dry biomass per block; mean ± SD) from non-mussel areas, 10 blocks containing small mussel aggregations (~20 mussels), 20 blocks of intermediate-size mussel aggregations (~50 mussels), and 10 blocks of large mussel aggregations (~80 mussels). All marsh blocks were transported back to the lab where they were washed completely clean

of all surface sediment. With the exception of 10 non-mussel blocks and 10 intermediate-size mussel aggregation blocks, all cordgrass was clipped to the marsh surface. For the 1-mussel treatments, we harvested 10 mussels (6–8 cm in length) from the experimental site and individually inserted them in the center of the marsh block so that they were 40–50% below the marsh surface.

After cleaning and cordgrass removal, all blocks were cut to new dimensions (36 cm x 36 cm x 16 cm) and placed within plastic-encased bins of the same dimensions. Bins containing marsh blocks were then centrally placed and fitted within an additional larger bin (61 cm x 61 cm x 8 cm), with the top of each box flush to the same height. The outside bin was filled with 64, 5 cm diameter PVC poles and 32, 2.5 cm diameter PVC poles (both 8 cm in height) so that all bin edges were held upright and PVC was rigidly filling all space within the outer box (Fig. S4). PVC poles were oriented in this way to capture all deposited sediment and minimize resuspension by substantially decreasing the fetch within the catchment bins. These sediment catchment units were then transported back to the experimental site where recipient holes were dug to the exact dimensions, so that the top of the marsh block (along with the top of each PVC pole) was exactly flush with the marsh surface sediment. We stapled 1-cm hardware cloth mesh (66 cm x 66 cm, with central 36 cm x 36 cm cutout) above PVC and flush to the marsh surface to allow invertebrate access to and from mussel aggregations and to limit the amount of disturbance to and resuspension of the settled material. Finally, to minimize mussel mortality in the absence of cordgrass, we built shades using 2 layers of 5-cm Aquamesh, attached these shades to four bamboo stakes, and inserted them above each plot at a height of ~1 m. The experiment ran for one month, from July 18 to August 18, 2017.

After one month in the field, all experimental units and their contents were returned to the lab, rinsed into recipient aluminum tins, dried, and weighed. The contents of the central bins and sediment on plant tissue were dislodged and collected using spatulas, scraper tools, and a Waterpik Flosser device. After all sediment was collected, each mussel was removed from the aggregation, measured for length, and weighed for biomass. Finally, from treatments containing vegetation, all aboveground cordgrass biomass was harvested, dried, and weighed (Fig. S6).

## Delft3D Model

To evaluate the contribution of mussel mounds to marsh accretion, we performed numerical simulations using the Delft3D-FLOW model[63,64]. We first modified the source code by adding a bivalve module (Delft3D-BIVALVES) to simulate sediment filtration and deposition processes that lead to mussel mound formations. In building this module, we assumed that mussels remove sediments from the water column because of filtration, and expel them as very cohesive pseudofeces, which are attached to the mounds, increasing their elevation. These processes are simulated by adding, in the computational cells containing the mussel mounds, a depositional term due to mussel filtration that reads:

$$\triangle z_{FILT} = \rho_{MM} \cdot f_{MM} \cdot C_{sed} \cdot dt \cdot \rho_{sed,dry}{}^{-1}, \qquad (2)$$

where $\rho_{MM}$ is the density of mussels in the mounds [mussel m$^{-2}$], set equal to 177 mussel m$^{-2}$[14], and $f_{MM}$ is the volume of water filtered by each mussel per unit of time [m$^3$ s$^{-1}$ mussel$^{-1}$], set equal to 0.115 m$^3$ s$^{-1}$ mussel$^{-1}$. $C_{sed}$ is the sediment concentration in the water column above each mussel mound [kg m$^{-3}$], $dt$ is the simulation time step [s], set equal to 0.6 s, and $\rho_{sed,dry}$ is the dry density of the sediments [kg m$^{-3}$], set equal to 800 kg m$^{-3}$[73]. The volume of sediments correspondent to the mussel filtration depositional term obtained from Eq 2. is removed from the lower computational layer of the water column above the mussel aggregation by adding the following sink term in the advection-diffusion equation:

$$SINK = \rho_{MM} \cdot f_{MM} \cdot C_{sed} \cdot A_{cell}, \qquad (3)$$

where $A_{cell}$ is the area of the computational cell [m$^2$]. Numerically, the term is implemented implicitly to prevent the appearance of negative concentrations. For settling velocity, we used a value of 0.1 mm s$^{-1}$. This value provides the best fit of the Total Suspended Sediment (TSS) concentration we surveyed in a creek, on the adjacent Little Sapelo Island, with an error of 0.022 ± 0.025 kg m$^{-3}$ (Fig. S8, MAE + RMSE). The fit was obtained by using the exponential decay formulation that reads:

$$C_s = C_{s0}e^{-t \cdot w_s/h}, \qquad (4)$$

where $w_s$ is the settling velocity in [m s$^{-1}$], $h$ is the slow depth in [m], $C_{s0}$ is the initial sediment concentration in [kg m$^{-3}$], and $t$ is the time in [s]. We set $C_{s0}$ equal to 0.10 g m$^{-3}$, which approximates the average value measured during flood tide, at the same location and tidal cycle. In addition, we set h equal to 0.30 m, which is the local mean annual high tide, calculated for 2018. To assess the sensitivity of the results to settling velocity, we ran a simulation in which we increased settling velocity by 50% (i.e., settling velocity equal to 0.15 mm s$^{-1}$), and extra deposition due to mussel mounds varied by only approximately 6.5% of the original value.

We next established a rectangular model domain to describe our study area in a simplified fashion (Fig. 5a). Within the model domain, the marsh platform is connected to the main channel by a tidal creek. The domain extends for 50 m and 207 m in the long-shore and landward directions, respectively. It is discretized using a rectangular grid constituted of 50 cm × 50 cm cells at the creek head and 50 cm × 100 cm cells elsewhere. In our model domain, mussel aggregations occupy only the creekhead, which is the 50 m × 50 m area between the creek and the upper part of the domain. We assign that each mussel mound has an area of 0.25 m$^2$, corresponding to a mound diameter of ~0.5 m. At our resolution, a mound occupies a single cell. A sensitivity analysis using cells of 0.25 m and 0.125 m showed negligible changes in the results. The main channel occupies the lower 20 m of the domain, and its depth goes from 0 m AMSL at the marsh edge to −6 m at the seaward boundary. The tidal creek is located in the middle of the marsh platform and stops 50 m from the landward boundary of the domain. It is 2 m wide, and its depth goes from 0.79 m AMSL at the creek head to −1 m where it connects to the main channel. The marsh system consists of four subareas: (i) the levees (0.94 m AMSL), which are 5 m wide cordons separating the marsh platform from the channel and the creek (except at the creek head) and are vegetated by tall-form cordgrass; (ii) the levee adjacent areas (0.79 m AMSL), which are 10 m wide and vegetated by intermediate size cordgrass, (iii) mussel aggregations, which occupy a set proportion of the creek head (0, 10, or 20%), are vegetated by short-form cordgrass, and form a regular array (0.79 m AMSL, a newly formed mound); and (iv) the marsh platform, all remaining area consisting of short-form cordgrass and located at a uniform elevation of 0.79 m AMSL (Table S2).

We used the Delft3D "trachytopes" functionality to impose vegetation resistance on flow propagating through the model domain. At every time step, a Chézy friction coefficient ($C$) is calculated for the vegetation, using a formulation developed by[83]. The formula is based on the unvegetated bed roughness ($C_b$), the drag coefficient ($C_D$), the vegetation height ($h_v$), and the vegetation density ($n$), expressed as the number of stems per square meter ($m$) times the stem diameter ($D_s$). In our model, only cells with an elevation higher than 0 m above MSL are vegetated. We considered four vegetation zones, as described above (Table S2; see details for collection of cordgrass and mussel parameters below). For each vegetation type, we used the same $C_b$ and $C_D$, equal to 45 m$^{1/2}$s$^{-1}$ and 1.65, respectively[84]. The vegetation properties, for each class, are based on local surveys and are reported in Table S1.

For each of the three mussel scenarios analyzed, we considered two vegetation distributions. The first one sticks with the description of the vegetation zones we report above. In the second scenario, the vegetation is absent from the entire domain.

To compute the sediment deposition in our numerical model, we simulated deposition from October 6th to October 22nd, 2018. This period contains the most representative spring and neap tides of the year and was obtained using the following procedure. First, we reconstructed the astronomic signal for 2018 using the tidal components of the NOAA station "Daymark #156, Head of Mud River, GA" # 8674975", which is the closest to our study area. We then calculated the tidal ranges in 2018 using consecutive low and high tide levels extrapolated from the astronomic tidal signal. Next, we classified the tidal ranges using the 25th and 75th quantiles of their distribution (i.e., Q25 and Q75): ranges lower than the 25th quartile were neap tides and ranges greater than the 75th were spring tides. The 2018 astronomic tide was then divided into periods containing a spring and a consecutive neap tide. For each period, we identified the tidal ranges associated with spring and neap tides by using Q25 and Q75. Finally, for each period, we calculated the average tidal range for neap and spring tide, the difference between these average values and the yearly average, and the sum of these two differences. The period with the lowest value of this sum contains the most representative spring and neap tides of 2018. For this date range, we then ran our model under six scenarios: mussel cover at 0, 10, and 20%, but with and without vegetation present. We report both sediment deposition and annual accretion in the five location types (i.e., levee crest, levee-adjacent, mussel aggregation, aggregation-adjacent, and non-mussel marsh platform) at local ($1\,m^2$), creekhead ($2500\,m^2$) and entire domain scales ($10,350$-$m^2$).

## Field experiment 3: creekshed mussel manipulation

To assess the effects of mussel presence and population size on marsh accretion at the creekshed scale, we first selected a marsh creekshed with three adjacent tidal creeks of similar length, structure, associated mussel populations, and marsh platform characteristics (i.e., size, elevation, and cordgrass characteristics). For each of the three tidal creeks, we first delineated a 50 m by 50 m creekhead area, oriented perpendicular to the direction of the tidal creek entry into the marsh, and located with midpoint of the front edge positioned at the point of tidal creek entry into the marsh. We then delineated a larger creekshed area associated with each creek of ≥10,000 $m^2$ within which we would deploy our experimental treatments. To quantify initial mussel and cordgrass cover, we set up three 50 $m^2$ transects (50 m long, 1 m wide) within the creekhead area, located at 0 m, 20 m, and 40 m distance from the tidal creek point of entry (and oriented perpendicular to the direction of flow). Within each transect, we counted each mussel aggregation, scoring each individual mussel as well as the length, width, and height from marsh platform of each mussel aggregation structure.

For a subset of 20 mussel aggregations randomly selected within each transect (3 transects per creek, 180 aggregations total), we scored the total number of cordgrass tillers on each aggregation. For a subset of 5 randomly selected tillers on each aggregation, we measured both length and width. To assess the differences in cordgrass characteristics between mussel aggregations and aggregation-adjacent areas, we also measured cordgrass stem density, height, and diameter in non-aggregation areas (1 $m^2$) located 1m away from each mussel aggregation.

After all initial data was collected, we removed and transplanted approximately 200,000 mussels from one tidal creekhead to another. To do so, we initially flagged approximately 4000 mussel aggregations within the creekshed of the "Removal" creek, encompassing both the 2500 $m^2$ creekhead area as well as the surrounding ≥10,000 $m^2$ creekshed extent. Mussel individuals were removed by hand over the

course of 16 weeks, with all field personnel taking care to leave all pseudofeces in place and cordgrass intact. Field crews were split between the mussel removal and mussel addition creek, such that mussels were re-transplanted within 24 h of removal to minimize mortality. Due to logistical and permitting constraints, it was not feasible to replicate the treatments across multiple sites; instead, the three plots occupied a single contiguous creekshed (Fig. 6a, b).

To assess changes in marsh elevation, we first quantified initial creekhead elevation (mean m AMSL in 2500-$ft^2$ area perpendicular to point of entry) using two metrics: 1) Real Time Kinematic (RTK) elevation datapoints (Trimble R6 GNSS System) distributed across the creekshed; and 2) measurements of mussel mound heights throughout each transect at set distances from the point of water entry. For the RTK datapoints, we collected 86 total points across the creekshed in June 2017. Elevation datapoints were randomly selected in each 2500 $m^2$ creekhead zone (minimum of 20 points per creekhead; Fig. S7). However, given the low number of RTK points across a large area, we additionally utilized mussel mound height calculations to provide a second estimate of initial elevation across the creekshed. Mussel aggregations and other bivalves, such as oysters, exhibit a height ceiling of growth, above which survivorship and growth are hypothesized to decrease. Previous work on Sapelo Island marshes reported the height ceiling to be +0.84 ± 0.004 m AMSL (mean ± SE). Therefore, assuming mature mussel aggregations (i.e., with tops at the aforementioned height ceiling), then mussel aggregation height (i.e., the distance between the marsh platform and the topmost point of the mussel aggregation mounded structure) will inform our knowledge of the marsh platform elevation by the following equation: Marsh Elevation (m AMSL) = Mussel Height Ceiling (+0.84 m AMSL) - Mussel Aggregation Height (m AMSL). For each distance from creekhead from which we conducted a 50 $m^2$ transect (0, 20, and 40 m), we estimated mean platform elevation using each of the measured mussel aggregation heights. We then took the mean value of marsh elevation across the three distances (0 m, 20 m, and 40 m) as a measure of creekhead elevation in 2017 for each of our experimental creeks (>60 mounds per creekhead; 250 total).

To assess elevation three years after treatment deployment, we compared creekhead elevation using a 2020 Digital Elevation Model (DEM) of the creekshed. To build the DEM, we flew a DJI Matrice 600 Pro drone carrying a custom build Lidar payload in August 2020. The payload consisted of a Velodyne Puck Lite VLP16, paired with a Novatel Stim300 Inertial Measurement Unit. The point clouds from the drone were orthorectified from GPS data continuously measured on the drone (see the procedure described in [85,86]). To remove the vegetation and any other surface perturbations (i.e., from digital surface model to digital elevation model), we used the CloudCompare software (https://github.com/cloudcompare/cloudcompare). The cloth Simulation Filter (CSF; [87]) was applied twice to the dataset, which successfully removed the vegetation data. The point cloud of the marsh surface was then exported to ArcGIS 10.7 where the DEM was generated by raster interpolation. Once completed, the mean elevation within each 2500 $m^2$ creekhead location was calculated using the Zonal Statistics tool in ArcGIS 10.7.

## Statistical analyses

To quantify the effects of season, tidal phase, and location type on short-term deposition, we first square root transformed short-term sediment deposition (i.e., filter paper results) to meet the assumptions of parametric statistics. We then conducted a three-way fully factorial ANOVA, with main effects season, tidal phase, and location type. Post-hoc analyses were conducted with Tukey HSD test, with Bonferroni-corrected $p$-values (STATA v 15.1). We further analyzed the effects of site, season, tide, and marsh location on short-term sediment deposition using regression tree analysis (rpart, R version 3.1.0). Over-fitted trees were pruned using k fold cross-validation. To next assess the

effects of marsh location type on total organic material deposited over 24 h (surface) and percent organic material (surface and to 5 cm depth), we ran three separate one-way ANOVAs. Post-hoc analyses were again conducted with Tukey HSD tests, with Bonferroni-corrected $p$-values (STATA v 15.1). For Experiment 1, we assessed the fate of mussel biodeposits, both previously settled and newly ejected, with a one-way ANOVA with location (creekhead versus platform) as the main effect. Finally, for Experiment 2, to assess whether cordgrass and mussel aggregations significantly affected sediment deposition over the one-month experimental deployment, we used multiple regression analysis with cordgrass biomass and mussel biomass as predictor variables for sediment biomass collected in each zone (STATA v 15.1; Table S1).

### Reporting summary

Further information on research design is available in the Nature Portfolio Reporting Summary linked to this article.

### Data availability

The data generated in this study have been deposited in the Figshare database under accession code: https://doi.org/10.6084/m9.figshare. 13177100.v4.

### Code availability

The code generated and/or utilized during the current study is available from the corresponding author on reasonable request.

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

## Acknowledgements

We thank A. Cetta and K. Prince for field support and A. Ortega for collecting the 2020 DEM. This work was supported by NSF CAREER Grant 1652628 to CA, NSF OCE 1832178 Award for the GCE LTER (UGAMI Contribution # 1112), University of Florida Seed Grant to CA, and NSF Graduate Research Fellowship Program Grant 1315138 to SMC. Logistical support was provided by Georgia Coastal Ecosystems Long Term Ecological Research Station, the University of Georgia Marine Institute, and the Sapelo Island National Estuarine Research Reserve.

## Author contributions

S.C., C.A., and T.B. conceived the study. S.C., H.F., and S.W. deployed field experiments. S.C. conducted landscape measurements, data collection, and statistical analysis of field data. D.P. and A.C. conducted the Delft3D-BIVALVE model. C.O. and N.R.D. contributed data. S.C. wrote the first draft of the manuscript. C.A., A.C., and D.P. edited the manuscript.

## Competing interests

The authors declare no competing interests.
