## [Peer Review File · Nature Communications]

Reviewer comments, first round -

Reviewer #1 (Remarks to the Author):

This paper focuses on the impact of mussels on marsh accretion. The overall subject of this paper, the role of fauna in marsh morphodynamics, is one that deserves more attention. The study combines field measurements, field experiments and modeling in what is clearly a significant research effort. Nevertheless, the results are somewhat limited in scope and location. The title and abstract indicate "southeastern US saltmarshes" (apparently comprising the Atlantic coasts of northern Florida, Georgia, and South Carolina) but the focus of the paper is on [mesotidal] saltmarshes at a site in Georgia (Sapelo Island) – many of their references regarding mussels are also based on results from Sapelo Island – with the implicit assumption that the results apply more generally to southeastern US saltmarshes. Statements about broader implications are mostly related to the general importance of faunal engineers rather than the particular results of this study.

The results convincingly show that mussels mounds located at the heads of tidal creeks locally increase deposition and accretion but almost no perspective is provided as to how much of total marsh area in southeastern US saltmarshes is occupied by tidal creekheads and how uniform the areal coverage and density of mussels is in these locations. Even at the study site, understanding the geomorphic context – both at the mound level and more broadly - was difficult without looking at the figures in Ref 14 (Crotty and Angelini 2020). The field results are used to develop an exploratory model using Delft3d with a new bivalve module to represent enhanced deposition related to bivalve filtering. The modeling provides some perspective on the relative contributions of mussel-mound enhanced deposition vs regular tidal deposition, but surprisingly does not include sediment redistribution, which is the focus of field experiment 1 and an important part of the paper's argument/hypothesis that locally produced biodeposits are redistributed more broadly over the marsh platform.

The paper is generally well written, but the pieces of the study, especially the modeling, did not seem well integrated. In addition, the format of having Methods at the end did not serve this paper well, as evaluating many of the results required understanding details of the methods. The field methods and statistics met the expected standards in the field, and with a few exceptions noted below, sufficient detail was provided in the methods for the work to be reproduced. Some points (conclusions) made at the beginning of the discussion seem overstated. Details are in comments below.

Specific comments:

L89-93: It would be appropriate to cite some of the seminal work of Jumars et al. on benthic fauna-sediment interactions.

L138-144: It would be helpful to use color to add information to Fig. 1, such as where the mussel aggregations are. Is this a mesotidal environment? How dependent are the results on the site being mesotidal? What are SSC values in tidal creek, rates of sea-level rise, elevation of marsh platform, marsh deposition? An inset map in Fig. 1 of the location of the study area would also be helpful.

L148: I know it is a requirement to put Methods at the end but without knowing more about how the filters were deployed, it is hard to know what to make of these results.

L159: Is this because mussel density and maximum spring tidal elevations vary seasonally?

L166-168: These numbers are only useful in a relative sense since the size of the filters not noted here (but provided in Methods). Better to normalize to filter area?

L168: It would be particularly helpful to provide some perspective on the deposition rates on the marsh platform at this site. Are there SETs or radiometric dating to give short or long-term deposition rates? Are these values comparable?

L176-178: This seems like an odd place to introduce a hypothesis. It would be better in the introduction and used to motivate field experiment 1.

L179: Was this done on a spring tide?

L185-193: Is this intended to suggest that these redistribution rates persist across additional tidal cycles? Have other marsh studies in this system found significant flow-associated transport on the marsh platform? This runs counter to many studies indicating limited redistribution on vegetated marsh platforms. Or is the redistribution facilitated by other fauna such as crabs?

L210: Was tidal submergence 2x longer at the creekhead site?

L238, Fig. 3: Is the width of the model (50 m) representative of the spacing of tidal creeks? If not, then the "entire marsh domain" deposition and accretion is representative of the tidal creek landscape but not the full marsh landscape.

L242: How long were the simulations? What tidal conditions? I see the Methods answer these questions but it is difficult to understand the results without knowing more about the simulations.

L249-251: These results suggest the effects of mussel-related enhanced deposition are very local and that sediment is not broadly redistributed through the system. I learned on L291-292 that the model "assumes no conveyance of biodeposits across marsh landscapes" [not noted in the Methods that I could see]. Why? If redistribution is central to the point of the paper, why not include redistribution in the model? It seems important in understanding the potential of hydrodynamic factors to disperse the biodeposits.

L254: Is -0.01cm/y the accretion rate for the creekhead and -0.08cm/y the accretion rate for landscape or is this a range of values for both related to scenario? If the former, why would deposition vary between sites by a factor of 2 and accretion by a factor of 8?

L258 and Fig. 3D,E: I am having trouble matching up values in text with values in the figure.

L260-262: Nice result.

L280: "2020 vegetation-filtered DEMs" is a little confusing, without having read the Methods. Something like "DEMs developed from drone imagery in 2020, filtered to remove vegetation, ..."

L284: How do these results compare with values from filters?

L287-289: Do mussels affect the geotechnical properties of the sediment or the presence of other fauna that could affect erosion rates when they are removed?

L292: Does the model in any way capture the rugosity of the mussel mounds and the potential for enhanced deposition simply by trapping sediment and preventing resuspension? Were any experiments with nonliving mussel mound structures carried out to see how much the structure itself contributes to enhanced deposition?

L301: Needs rewording: "of particularly of inorganic material"

L302: The results do not convince me that the enhanced deposition is "redistributed across marsh platforms", which implies to me something more than local redistribution.

L304-305: This statement also seems too strong. There is nothing in the results about SLR, which may affect the system in a multitude of ways that could impact the role of biodeposition.

E.g., can localized high deposition create drainage issues that contribute to pond formation?

The hydrodynamic aspects of this seem like something the model could address.

L313, Fig. 5: It is not clear why fauna other than mussels are being brought into the picture here. This would be more appropriate for a review-type article on engineering fauna.

L320: Should this sentence about indirect effects follow the text on L314-317 about indirect effects?

L324: To be convinced that the contribution of mussels to sediment deposition over space and time can be predicted, I'd have to be convinced that the distribution and density of mussels can be predicted. The paper doesn't address that. In fact, it is noted as necessary next step in L345.

L351: "coastal ecosystems"

L363: It is hard to visualize the way the filters are deployed. How is the wire used to attach the filter to the petri dish? How is the petri dish affixed to the marsh or mussel aggregations. Is the petri dish level? How high is the lip on the petri dish? Perhaps a photo could be added to the Supplementary figures to show how this was done.

L373: These filters were deployed without petri dishes?

L381: Needs rewording: "in persistent in marsh"

L426: Regarding the marsh blocks being "washed completely clear of all surface sediment", is there no surface sediment on the marsh platform? Can this be done without disturbing sediment below the surface? It is hard to imagine what the marsh blocks looked like after cleaning.

L459: How sensitive are the results to the density of mussels in a mound? Do all mussel mounds found on creekbanks have a similar number of mussels? Do they all filter a similar amount? Does it vary by life stage, temperature, other factors?

L464: I don't think tidal creek SSC is ever noted in the paper, so the error is hard to evaluate. It would be helpful if tidal creek SSC at different tidal stages was provided. How sensitive are the

results to the value of settling velocity?

L474: Is the overall area covered by mussels held constant in the sensitivity analysis or is it varied? Are the results sensitive to the total area of mussels?

L540: This must have entailed many trips across the marsh. How was disturbance to the marsh surface minimized?

L558: "did inform"?

L826: Isn't this the same as Fig. 5A?

Random question: Do mussels inhibit or accelerate headward creek extension?

Reviewer #2 (Remarks to the Author):

I have reviewed the manuscript entitled 'Faunal engineering stimulates landscape-scale accretion in southeastern US salt marshes' (NCOMMS-22-26048). This manuscript describes a series of five aligned experiments and modelling efforts which together assess the scale and density dependent role of mussel aggregations in marsh sediment deposition and accretion. The manuscript represents a novel combination of field data and spatial models and I particularly applaud the authors for their investigation that spans multiple spatial scales which are critical to inferring real-world effects of organisms on coastal ecosystem function and geomorphology. The manuscript presents substantial evidence for the role of mussel aggregations in promoting marsh accretion both at local site and estuary wide scales and a variety of indirect and direct effects of mussels are attributed to this role.

The results are an important addition to the literature and interesting. However, I have a few suggestions to improve readability, understanding, and interest:

Sea-level rise linkage: I find the linkage between the pace of sea-level rise and rates of accretion are interesting and an important implication of this work. However, the linkages as they are stated in the manuscript are tenuous without some further exploration of the assumptions of sea-level rise changes to marshes – for example, sea-level rise is unlikely to solely change water depth, it is likely to change hydrodynamic regimes and therefore there will probably be interactions with rates of accretion in marshes. Perhaps in the discussion, the authors could add a little more nuance to the linkage between their results and the implications for changes under various sea-level rise projections.

Results and clarification: It is a challenge and an art to present results in the stand alone format required for this journal and compounding that, this paper includes methods and results for five different studies that could all form papers on their own. I see a lot of merit in presenting the findings from all five together as it builds clear evidence for the profound effects of mussels on coastal geomorphology and function at multiple scales and contexts. However, as is, the results are difficult to follow because a lot of the methodological detail is buried either in the supplement or the methods at the end. These issues may be able to be resolved with some edits to clarify (see below for detailed comments). But arguably, the manuscript perhaps tries to fit too much detail for each experiment/survey/model, and it could be best to focus on one key finding from each study that contributes evidence to the overarching multi-scale story. The authors might also consider one overarching conceptual figure to tie all five studies together, which could be a map that has the scale of each experiment/model marked and the key findings summarized for each. This would also emphasize the novelty of this approach.

Some examples where clarification is required:

- The terminology to describe the sampling positions is confusing to follow in the text (e.g., outer-marsh levee, levee, levee crest, levee-adjacent, non-mussel high marsh platform, aggregation-adjacent, etc.), and it also appears that the terminology is not consistent throughout the manuscript (e.g., compare position descriptions on lines 142-151 with 354-361). This may seem like a minor issue, but it makes the results difficult to follow and interpret.

- Figure 1 – I am not sure how to interpret this figure – for example, the caption states that the bars with black borders are the mussel aggregation data, but what are the bars without black

borders around them? The scale bar has two colors, but what do those colors represent because the bars in the figure are only one color? Perhaps also label all the position descriptors that are used in the text here?

- In places, there is not enough methodological information in the results to interpret the figures, and some important details are buried in the supplementary material; for example, Figure 2C shows a dotted line for the 'open control' results, but it is only when I look at the supplementary Figure 3 that I can see what an 'open control' is. Further, the open controls have no mussels, but the line spans the mussel gradient on the x axis which is confusing.

- Experiment 2 – the written text on line 203-217 refers to the different positions as 'low' and 'high' tidal elevations, but in Figure 2 and in other places in the text they are referred to as 'Creekhead' and 'Platform'. This is another example of inconsistency in terminology which makes the results difficult to follow.

- In Figure 2C – which relationships do the linear lines correspond to? The relationships with or without cordgrass?

- Line 280 – Acronym DEMs is not defined earlier

- Line 200-202 and 417-421 – Instead of 'intermediate size mussel aggregations plus no cordgrass', could you simplify treatment names to a series of "+mussels/-cordgrass" treatment labels? E.g., +++mussels/-cordgrass (80 mussels and 0 cordgrass)? This naming system would improve ability to follow the different treatment effects in the written text.

- Line 440 – A 1 cm mesh cloth was attached to the sediment trap boxes, but the manuscript doesn't state why this mesh is required.

Reviewer #3 (Remarks to the Author):

This is an exciting and creative study with surprising results, highlighting the importance of mussels on marsh accretion. However, I do have a number concerns before I can recommend its publication.

One particular concern is the missing vegetation effect on sediment deposition in both your experimental work and the modeling. If I did not misunderstand, you even demonstrate negative effects of vegetation presence on sedimentation. This clearly contradicts with the available experimental and modeling literature. How do you explain this? Can you be certain that your results on mussel effects are robust when your experiments and your modeling cannot replicate these well-established biogeomorphic feedbacks?

Along the same lines, I miss data on the vegetation density (or structure, or standing biomass). I did not find any data on this. How do they relate to field data or those presented in other mesocosm experiments (i.e. marsh organs)?

I attach a pdf with some additional comments.

REVIEWER COMMENTS

Reviewer #1 (Remarks to the Author):

Comment 1-1: This paper focuses on the impact of mussels on marsh accretion. The overall subject of this paper, the role of fauna in marsh morphodynamics, is one that deserves more attention. The study combines field measurements, field experiments and modeling in what is clearly a significant research effort. Nevertheless, the results are somewhat limited in scope and location. The title and abstract indicate “southeastern US saltmarshes” (apparently comprising the Atlantic coasts of northern Florida, Georgia, and South Carolina) but the focus of the paper is on [mesotidal] saltmarshes at a site in Georgia (Sapelo Island) – many of their references regarding mussels are also based on results from Sapelo Island – with the implicit assumption that the results apply more generally to southeastern US saltmarshes. Statements about broader implications are mostly related to the general importance of faunal engineers rather than the particular results of this study.

The results convincingly show that mussels mounds located at the heads of tidal creeks locally increase deposition and accretion but almost no perspective is provided as to how much of total marsh area in southeastern US saltmarshes is occupied by tidal creekheads and how uniform the areal coverage and density of mussels is in these locations. Even at the study site, understanding the geomorphic context – both at the mound level and more broadly - was difficult without looking at the figures in Ref 14 (Crotty and Angelini 2020).

We thank the reviewer for these comments and we agree that data on both the regional and geomorphological context was lacking in the previous version of this manuscript.

To address the regional context, we compiled new data on creekhead density, mussel coverage, and total areal coverage of both creekheads and mussel aggregations throughout the region. We show through new statistical analysis that Sapelo Island is representative of the broader region, spanning northeast Florida to South Carolina. We paste the new findings which are reported in the Results section and figure below.

Results, Lines 143-165: The South Atlantic Bight (SAB) region, extending from Cape Fear (NC) to Cape Canaveral (FL), contains the most expansive salt marsh systems in the U.S.—totaling approximately 750,000 acres. The outer margins of the SAB are microtidal, but they quickly become mesotidal approaching the head of the Bight (55). River discharge contributes a substantial and continuous supply of suspended fine grain sediment to support vertical accretion of these salt marshes (56). To first characterize tidal creekhead density across the SAB, we quantified the total number of tidal creekheads in 1km² marsh areas at 10 sites spanning >200 miles of coastline from Cape Romain, South Carolina to Amelia Island, Florida. Creekhead density was 110.2 ± 6.3 creeks km⁻² (mean ± SD, here and below) and there were no significant differences between the northern (106.5 ± 19.7 creeks km⁻²), Sapelo Island (study site; 98.3 ± 13.1 creeks km⁻²), and southern sites (127.0 ± 13.9 creeks km⁻²; Figure 2A, B; p>0.05). Given that the creekhead mussel ‘halo’ extends approximately 50m into the marsh (14), we define creekhead area of influence to be a 50m x 50m area oriented perpendicular to the creek and with the midpoint of one side at the main point of creek entry onto the marsh. Based on this assumption, we calculate creekhead area to be 27.6 ± 5.0% of the total marsh area in this region (Figure 2C, light gray bars).

To next characterize mussel coverage across the SAB, we surveyed 12 total sites spanning >150 miles of coastline from Edisto Beach, South Carolina to Amelia Island, Florida. Throughout the SAB, mussel aggregations comprise 10.3 ± 2.1% (mean ± SD, here and below) of total creekhead area (Figure 2A, B). There were no significant differences in mussel coverage of tidal creekheads across the northern (9.3 ± 0.8%), Sapelo Island (11.3 ± 1.0%), and southern sites (9.6 ± 3.2%; p>0.05). When scaled to the marsh creekshed (i.e., not simply tidal creekheads), we calculate mussel areal coverage to be 2.8 ± 0.6% of the total marsh area in the region (Figure 2C). Based on these results, we suggest that our study site, Sapelo Island, is a salt marsh system representative of the broader SAB region with regards to creekhead density, creekhead area, and mussel cover at the landscape scale.

Methods, Lines 400-428: To understand the variation in salt marsh geomorphology and mussel coverage across the South Atlantic Bight (SAB), we assessed density and areal coverage of (1) tidal creekheads and (2) mussel aggregations with a combination of published data and new field surveys across the region. First, to assess creekhead density, we selected 10 sites ranging from Cape Romain (SC) to Amelia Island (FL). Given that all of our experiments were conducted on Sapelo Island and the surrounding marsh islands, we selected three sites on Sapelo Island for comparison with four sites to the north and three to the south (61). At each site, we scored the total number of tidal creekheads in a 1-km² contiguous marsh area using Google Earth. Next, assuming each tidal creekhead constitutes approximately 0.0025km², we calculate the creekhead areal coverage to be:

$$\text{Creekhead Areal Coverage (\%)} = \frac{(0.0025\text{km}^2) \times \text{Creekhead Density (\# km}^{-2}\text{)}}{\text{Marsh Creekshed Area (1km}^2\text{)}} \times 100\%$$

Differences across northern, Sapelo Island, and southern sites were assessed with a one-way ANOVA with location as the main factor.

To next test the hypothesis that creekhead mussel coverage is similar at sites across the SAB, we conducted surveys of mussel aggregations at 12 sites across the region from Edisto Beach (SC) to Amelia Island (FL). Previous work (14) has shown that mussel aggregations decrease in size and density with increasing distance from the tidal creekhead, so we focused our measurements at three distances from one tidal

creekhead onto the marsh platform: 0m, 20m, and 40m. We note that at all sites, mussel aggregations extended >40m from the tidal creekhead. Sites were again distributed across the region, and included 3 sites to the north, 3 sites to the south, and 6 sites on Sapelo and its back barrier marsh islands. At each site, we selected one representative creek 100-175m in length and ensured that the tidal creekhead did not overlap spatially with a tidal creekhead of an adjacent creek. At each distance from creekhead, we established one 50m x 1m transect. Walking the transect line, we scored each mussel aggregation, counting the total number of mussels and measuring the mound dimensions (LxWxH). We then calculated the areal coverage of mussels within each transect (50m²) and took the mean value across the three distances as the measure for the site. All data was collected between May and August in 2016 and 2017. Differences across northern, Sapelo Island, and southern sites were assessed with a one-way ANOVA with location as the main factor. Finally, we calculated creekshed mussel areal coverage in the three sub-regions, as the product of the percent of creekshed occupied by creekheads (sub-region mean, %) and the proportion of creekhead area occupied by mussels at each site.

Figure 2. Creekhead density and mussel areal coverage at sites distributed across the South Atlantic Bight. (A) Regional surveys quantified the percent area of tidal creekheads occupied by mussels (values on the left of Panel A with sites denoted in open circles) and the density of tidal creekheads within larger creekshed areas (# 1 km⁻²; values on the right of Panel A with sites denoted in black circles). (B) Sapelo Island sites were similarly assessed for creekhead mussel area (open circles) and creekhead density (black circles). (C) There were no statistically significant differences in creekhead density (light grey bars) or creekhead mussel area (dark gray bars) across northern sites, Sapelo Island sites, and southern sites (mean +SE). Finally, assuming a creekhead area of 2,500m², areal coverage of creekheads and of mussels range from 25-32% and 2.5-3.0%, respectively across the region (mean values reported to right of error bars).

By including a summary of these data which are derived from the authors' fairly extensive work across the region and adding a new section to the Results, ("Regional Context"), we hope to provide sufficient evidence that the results we provide from Sapelo Island and its surrounding marshes have relevance to the broader southeastern US region. Given how closely our sites match the geomorphic and ecological context of the region as confirmed by these new results, we suggest it is appropriate to retain the 'southeastern US' regional focus in the title.

To next address the geomorphic context, we now add a more explicit description of the hierarchy of factors controlling mussel population coverage (Lines 111-120), as well as a figure adapted from Crotty & Angelini (2020) to the Supplemental Information to aid in clarifying this concept.

Lines 111-120: In mesotidal salt marshes of the southeastern US, mussels aggregate in clusters of up to 200 individuals on marsh platforms, with individual aggregations commonly exceeding 1-m². Previous work (14)

elucidated a hierarchy of factors controlling mussel cover at the creekshed (i.e., creek length and associated tidal prism), landscape (i.e., elevation and the associated submergence time), and patch scale (i.e., predation regimes; see Figure S2). Findings revealed that these biogenic features predictably occur in the highest densities in close proximity to creekheads—where tidewater floods and ebbs from the marsh with each tide—and decrease in density with increasing distance from creekheads onto adjacent marsh platforms (14, 44, 52). Therefore, while mussels exhibit patchy spatial cover across landscapes, their presence and densities can be reliably predicted by easily-identifiable tidal creek features (14).

Figure S2. The hierarchy of geomorphology and species interactions that occur across the creekshed, landscape, and local patch scales that predictably control the presence and population size of mussel populations in southeastern US salt marshes (Figure adapted from Crotty and Angelini 2020). At the A) creekshed scale, longer tidal creeks with larger inlet cross-sectional areas support larger tidal prisms and higher creekshed recruitment and growth than shorter creeks of similar complexity. At the B) landscape scale, mussel aggregations in close proximity to the creek head are positioned at lower marsh platform elevations and experience higher water flows, recruitment, and growth than those located in the marsh interior. At the C) patch scale, mussels located on existing aggregations experience reduced predation and desiccation and have higher survivorship than mussels located off of existing mussel aggregations. The D) hierarchy of geomorphology and species interactions controlling mussel population recruitment, growth, and survival is organized by measured features, the physical/biological driver represented, and the mussel population effect (green coloring refers to creekshed scale, blue coloring refers to landscape scale, and yellow coloring refers to the patch scale).

Comment 1-2: The field results are used to develop an exploratory model using Delft3d with a new bivalve module to represent enhanced deposition related to bivalve filtering. The modeling provides some perspective

on the relative contributions of mussel-mound enhanced deposition vs regular tidal deposition, but surprisingly does not include sediment redistribution, which is the focus of field experiment 1 and an important part of the paper's argument/hypothesis that locally produced biodeposits are redistributed more broadly over the marsh platform.

We agree with the reviewer that the inclusion of sediment redistribution into the model will be an important factor to include to more accurately assess the fate of the material deposited on mussel aggregations. In the first iteration on this manuscript, we focused on modeling the contribution of mussel mounds to the volume of sediment deposition, and therefore, we did not consider resuspension or redistribution in the model. This decision was based on previous research indicated that the majority of sediments deposited on marsh platform do not get resuspended and exported (Christiansen et al., 2000; Green and Coco, 2014). We now suggest that based on results from Experiment 1, and the ability of bioturbating organisms that co-occur on mussel aggregations to resuspend sediment, that redistribution of sediment from mussel aggregations and into adjacent marsh areas is a key feature enabling mussels to continuously and more broadly contribute to marsh sediment budgets.

To better quantify the contribution of erodibility (i.e., ability to be resuspended and either transported across the marsh or out of the system) on the results, we now run a sensitivity analysis in which we allow for the resuspension of material previously deposited on mussel aggregations, setting a critical shear stress for erosion ($\tau_{c,e}$) equal to 1 Nm^{-2} (Mariotti and Canestrelli, 2017). This way, sediment deposited on the mussel mounds can be resuspended by the water fluxes, and consequently be transported to the marsh platform, or alternatively, be exported offshore. We find that, on one hand, resuspension increases sediment spread through the marsh as predicted. However, on the other hand, a small percentage of resuspended sediment exits the marsh through the creek. In either case, there is a significant impact of mussel mounds on sediment trapping in the marsh platform.

We now include the results of the sensitivity analysis in the text (see lines 291-297). We do not, however, change the full set of modeling scenarios. We elected for this path forward because we do not feel that the sensitivity analysis satisfactorily captures the differential resuspension on mussel aggregations versus on marsh platforms, nor does it capture dominance of flood versus ebb tides (and the co-occurrence of peak bioturbation). We suggest that the far greater effect of mussels on landscape accretion in Experiment 3 suggests that these factors may increase the contribution of mussels, rather than diminish them. Therefore, we feel that our initial model captures a lower bound estimate for mussel contribution, and we argue that future modeling work should address this key area of needed advancement.

T. Christiansen, P.L. Wiberg, T.G. Milligan, Flow and sediment transport on a tidal salt marsh surface. *Estuar. Coast. Shelf Sci.*, **50**, 315–331 (2000).

M.O. Green, G. Coco, Review of wave-driven sediment resuspension and transport in estuaries. *Rev. Geophys.*, **52**, 77-117 (2014).

G. Mariotti, A. Canestrelli, Long-term morphodynamics of muddy backbarrier basins: Fill in or empty out? *Water Resources Research*, **53**, 7029–7054 (2017).

Comment 1-3: The paper is generally well written, but the pieces of the study, especially the modeling, did not seem well integrated. In addition, the format of having Methods at the end did not serve this paper well, as evaluating many of the results required understanding details of the methods. The field methods and statistics met the expected standards in the field, and with a few exceptions noted below, sufficient detail was provided in the methods for the work to be reproduced. Some points (conclusions) made at the beginning of the discussion seem overstated. Details are in comments below.

We agree and apologize for the lack of clarity in the initial version of the paper. To address these concerns (and similar concerns of Reviewer 2), we have made the following changes:

- 1) Included an overarching conceptual figure outlining the spatial and temporal scale of all of our field components as well as associated hypotheses (see Figure 1, pasted below)

Figure 1. Conceptual figure outlining the spatial and temporal scale of all field components and associated hypotheses. (1A-C) Landscape assays of sediment deposition (i.e., 9-cm filter papers; 57) were distributed across 13 area types over four 24-hour tidal deployments. We hypothesized that sediment deposition atop mussel aggregations would be as high as deposition on levee crests. (2A,B) Experiment 1 involved tracking the fate of fluorescently tagged previously-settled and newly ejected biodeposits from mussel aggregations (24-hour

deployments). We hypothesized that sediment would be rapidly redistributed across marsh platforms from these local hot spots of deposition. (3A,B) Experiment 2 involved the deployment of seven treatments containing a range of mussel and cordgrass biomass. Treatments were deployed at the creekhead and on the marsh platform in sediment catchment devices, designed to capture all sediment deposited throughout the 1-month deployment. We hypothesized that mussel biomass would drive sediment deposition at this intermediate temporal and spatial scale. (4) Experiment 3 involved the removal of mussels from one tidal creekhead and the transplantation of these mussels to another proximate creekhead. We hypothesized that the removal of mussels inhibits accretion at the landscape scale, while addition increases it relative to an unmanipulated control. Locations of each experiment are highlighted in the panels at left. Numbers and colors correspond to the experiment of relevance.

- 2) Streamlined results to improve clarity
 - a. Provided the regional context at the beginning of the results allowing us to better define marsh area types (e.g., creekheads and mussel aggregations) up front.
 - b. Moved analysis of organic and inorganic sediment contribution to the supplement.
 - c. Removed additional regression tree analysis to simplify the translation of our findings.
 - d. Clarified Delft-3D model results and added five sensitivity analyses as recommended by reviewers.

- 3) Revised the results section to include key methodological statements wherever possible (e.g., Lines 169, 198-201, 208-213, etc.).

Specific comments:

Comment 1-4: L89-93: It would be appropriate to cite some of the seminal work of Jumars et al. on benthic fauna-sediment interactions.

We now cite this seminal work in lines 89-93.

Lines 89-93: Ecosystem engineering fauna, hereafter 'faunal engineers', can modify sediment deposition and accretion processes through a variety of direct and indirect mechanisms (e.g., 41). Suspension- and filter-feeding organisms, such as bivalves and sponges, actively contribute to sediment deposition, while bioturbating organisms, such as burrowing crabs and worms, resuspend sediment into the water column (41, 42).

41. PA Jumars, ARM Nowell, RFL Self, A simple model of flow-sediment-organism interaction. *Mar Geol* **42**:155-172 (1981).

42. Nowell ARM, Jumars PA, and Eckman JE (1981). Effects of Biological Activity on the Entrainment of Marine Sediments. *Mar Geol*, **42**:133-153.

Comment 1-5: L138-144: It would be helpful to use color to add information to Fig. 1, such as where the mussel aggregations are. Is this a mesotidal environment? How dependent are the results on the site being mesotidal? What are SSC values in tidal creek, rates of sea-level rise, elevation of marsh platform, marsh deposition? An inset map in Fig. 1 of the location of the study area would also be helpful.

We now add color to Figure 1 to highlight the locations of mussel aggregations and filter paper deployments. In addition, as part of the conceptual figure, we include an inset map of the locations of each field deployment.

Regarding the context of the environment, tides along the Georgia coast have an average amplitude of 2.4 m and a spring tide range of 3.4 m (Chalmers 1997). This mesotidal environment, and the hydraulic energy

associated with the tide, effects many of the ecological processes in the marshes (Schelske and Odum, 1961). Therefore, we agree that the results are likely only immediately generalizable to marshes that have similar flooding and draining regimes. We now provide greater regional context for mesotidal marshes.

Finally, while we do not have elevation measurements for each marsh location type or SSC in each creek, we do now include a comparison of filter paper results to a range of accretion rates collected from across the region in Crotty et al (2020). Please see Comment 1-9 (below) for more information.

Comment 1-6: L148: I know it is a requirement to put Methods at the end but without knowing more about how the filters were deployed, it is hard to know what to make of these results.

We understand this concern and agree. We now include a new Figure (see Comment 1-3) and take on suggested improvements to the previous version of Figure 1 (see Comment 1-5) to improve clarity and interpretability of this work. We also include a figure and additional methods in the supplement that provide further details about how the filters were deployed. See comment 1-31 below.

Comment 1-7: L159: Is this because mussel density and maximum spring tidal elevations vary seasonally?

With the exception of a period of time in early June when mussel larvae (which are highly vulnerable to post-settlement predation) settle in salt marshes in the region, mussel density does not substantially vary across seasons according to numerous field surveys and experiments we have conducted in these marshes. However, the reviewer is correct that the maximum spring high tides do vary somewhat across seasons with winter tides in December, January and February only reaching between +2.1 and +2.5 meters while those in late summer and fall reach a maximum of +2.6 meters. However, in our case the summer spring tide and winter spring tide reached the same height (+2.5m), and the neap tides simply differed by 0.1m across the seasons (+2.1m in summer and +2.0m in winter). Ultimately, it is likely that the low deposition on mussel mounds during winter spring and neap tides, relative to summer tides, is the result of shorter inundation times as well as the colder temperatures (and the resulting slower suspension feeding/metabolic activity).

Comment 1-8: L166-168: These numbers are only useful in a relative sense since the size of the filters not noted here (but provided in Methods). Better to normalize to filter area?

We agree that these numbers are relative. However, we elect to keep them in units of kg/m²/day because we report results from experiment 2 in the same units (see text and Figure 4). We retain these consistent units of measurement so that the reader can compare the magnitude recorded in these 24-hour deployments with those values recorded over a month. However, we do now include a mention of filter diameter in the results section (Line 169).

Comment 1-9: L168: It would be particularly helpful to provide some perspective on the deposition rates on the marsh platform at this site. Are there SETs or radiometric dating to give short or long-term deposition rates? Are these values comparable?

In our previous work, we compiled accretion data from across this region from a variety of methods (Crotty et al 2020; Supplemental Table pasted below). We now compare these values to filter paper deposition using the following steps which are now included in Supplemental Methods Section.

Butzeck et al (2015) propose the equation for converting sediment deposition rate (SDR; g cm⁻² yr⁻¹) into accretion rate (AR; cm yr⁻¹): $AR (cm\ year^{-1}) = SDR (g\ cm^{-2}\ year^{-1}) / BD (g\ cm^{-3})$, where BD is sediment bulk density in g cm⁻³. We collected bulk density measurements from the region (Bradley and Morris 1990) and estimated an average bulk density to be between 0.3 (Neubauer et al 2002) and 1 g cm⁻³ (Butzeck et al 2015).

We then compare measured filter paper deposition values of 0.02 (non-mound neap), 0.11 (non-mound spring), 0.08 (on mound neap) 0.39 (on mound spring), and 0.61 kg m⁻² day⁻¹ (summer on-mound spring). We calculate accretion rates as follows:

$$\text{Non-mound neap: } 0.02 \text{ kg m}^{-2} \text{ day}^{-1} * (365 \text{ days} / 1 \text{ yr}) * (1000 \text{ g} / 1 \text{ kg}) * (1 \text{ m}^3 / 1,000,000 \text{ cm}^3) = \\ 0.0073 \text{ g cm}^{-2} \text{ yr}^{-1} / 0.3 \text{ g cm}^{-3} = 0.024 \text{ cm yr}^{-1} \text{ or } 0.24 \text{ mm yr}^{-1}$$

$$0.0073 \text{ g cm}^{-2} \text{ yr}^{-1} / 1 \text{ g cm}^{-3} = 0.0073 \text{ cm yr}^{-1} \text{ or } 0.07 \text{ mm yr}^{-1}$$

$$\text{Non-mound spring: } 0.11 \text{ kg m}^{-2} \text{ day}^{-1} * (365 \text{ days} / 1 \text{ yr}) * (1000 \text{ g} / 1 \text{ kg}) * (1 \text{ m}^3 / 1,000,000 \text{ cm}^3) = \\ 0.040 \text{ g cm}^{-2} \text{ yr}^{-1} / 0.3 \text{ g cm}^{-3} = 0.133 \text{ cm yr}^{-1} \text{ or } 1.33 \text{ mm yr}^{-1}$$

$$0.040 \text{ g cm}^{-2} \text{ yr}^{-1} / 1 \text{ g cm}^{-3} = 0.040 \text{ cm yr}^{-1} \text{ or } 0.40 \text{ mm yr}^{-1}$$

$$\text{On-mound neap: } 0.08 \text{ kg m}^{-2} \text{ day}^{-1} * (365 \text{ days} / 1 \text{ yr}) * (1000 \text{ g} / 1 \text{ kg}) * (1 \text{ m}^3 / 1,000,000 \text{ cm}^3) = \\ 0.029 \text{ g cm}^{-2} \text{ yr}^{-1} / 0.3 \text{ g cm}^{-3} = 0.097 \text{ cm yr}^{-1} \text{ or } 0.97 \text{ mm yr}^{-1}$$

$$0.029 \text{ g cm}^{-2} \text{ yr}^{-1} / 1 \text{ g cm}^{-3} = 0.029 \text{ cm yr}^{-1} \text{ or } 0.29 \text{ mm yr}^{-1}$$

$$\text{On-mound spring: } 0.39 \text{ kg m}^{-2} \text{ day}^{-1} * (365 \text{ days} / 1 \text{ yr}) * (1000 \text{ g} / 1 \text{ kg}) * (1 \text{ m}^3 / 1,000,000 \text{ cm}^3) = \\ 0.142 \text{ g cm}^{-2} \text{ yr}^{-1} / 0.3 \text{ g cm}^{-3} = 0.473 \text{ cm yr}^{-1} \text{ or } 4.73 \text{ mm yr}^{-1}$$

$$0.142 \text{ g cm}^{-2} \text{ yr}^{-1} / 1 \text{ g cm}^{-3} = 0.142 \text{ cm yr}^{-1} \text{ or } 1.42 \text{ mm yr}^{-1}$$

$$\text{Summer On-mound spring: } 0.61 \text{ kg m}^{-2} \text{ day}^{-1} * (365 \text{ days} / 1 \text{ yr}) * (1000 \text{ g} / 1 \text{ kg}) * (1 \text{ m}^3 / 1,000,000 \text{ cm}^3) = \\ 0.222 \text{ g cm}^{-2} \text{ yr}^{-1} / 0.3 \text{ g cm}^{-3} = 0.74 \text{ cm yr}^{-1} \text{ or } 7.40 \text{ mm yr}^{-1}$$

$$0.222 \text{ g cm}^{-2} \text{ yr}^{-1} / 1 \text{ g cm}^{-3} = 0.222 \text{ cm yr}^{-1} \text{ or } 2.22 \text{ mm yr}^{-1}$$

Our measured rates of accretion range from +0.01 cm yr⁻¹ to +0.74 cm yr⁻¹. This range of accretion values is comparable to those found in the region (-0.4 to +1.0 cm yr⁻¹). We note that results from Experiment 2 were also in this same range, 0.04 - 0.40 kg m⁻² day⁻¹, equivalent to (at the low end) 0.015 cm yr⁻¹ and (at the high end) 0.487 cm yr⁻¹.

The main text now reads as follows:

Lines 181-184: We note that, when converted to accretion rates (60), the filter paper results range from +0.01 cm yr⁻¹ to +0.74 cm yr⁻¹, values comparable to those found in the region (-0.4 to +1.0 cm yr⁻¹) from surface elevation tables (SET), feldspar marker horizons, 137 Cs, and 210 Pb data points (61; see Supplemental Methods for calculations).

Citations (including for tables on next two pages):

- 1) Butzeck C, Eschenbach A, Grongroft A, and Hansen K. (2015). Sediment Deposition and Accretion Rates in Tidal Marshes Are Highly Variable Along Estuarine Salinity and Flooding Gradients. *Estuaries and Coasts*, 38, 434-450.
- 2) Bradley PM and Morris JT. (1990). Physical characteristics of salt marsh sediments: ecological implications. *Marine Ecology Progress Series*, 61, 245-252.
- 3) M.J. Loomis, C.B. Craft, Carbon Sequestration and Nutrient (Nitrogen, Phosphorus) Accumulation in River-Dominated Tidal Marshes, Georgia, USA. *Wetland Soils* 74,1028-1036 (2010).
- 4) C.R. Alexander, J.Y.S. Hodgson, J.A. Brandes, Sedimentary processes and products in a mesotidal salt marsh environment: insights from Groves Creek, Georgia. *Geo-Mar Letters* 37, 345-359 (2017).
- 5) K.B. Raposa et al., Assessing tidal marsh resilience to sea-level rise at broad geographic scales with multi-metric indices. *Biol Conserv* 204, 263-275 (2016).
- 6) P. Sharma, L.R. Gardner, W.S. Moore, M.S. Bollinger, Sedimentation and bioturbation in a salt marsh as revealed by 210Pb, 137 Cs, and 7 Be studies. *Limnol Oceanogr* 32, 313-326 (1987).
- 7) R.L. Vogel, B. Kjerfve, R. Gardner, Inorganic sediment budget for the North Inlet salt marsh, South Carolina U.S.A. *Mangroves and Salt Marshes* vol. 1, no. 1, pp 23-35. SPB Academic Publishing, Amsterdam, The Netherlands (1996).
- 8) J.T. Morris, P.V. Sundareshwar, C.T. Nietch, B. Kjerfve, D.R. Cahoon, Responses of coastal wetlands to rising sea level. *Ecology* 83, 2869-2877 (2002).

Table S1 from Crotty et al (2020). Salt marsh vertical accretion data site coordinates, methods, estimated vertical accretion rate and source. Data sources are presented both inset (rightmost column) and in full detail below table. Accretion rate is presented in units of mm per year.

STATE	LATITUDE	LONGITUDE	METHOD	YEAR CORE COLLECTED	YEARS OF SET	ELEV. (NAVD88)	MARSH ZONE	RATE (MM/YR)	SOURCE
GA	31°21'4.10"N	81°20'1.42"W	137 Cs	2001	1964-2001	NA	Platform	1.4	Loomis & Craft 2010
GA	31°23'14.80"N	81°16'50.39"W	137 Cs	2001	1964-2001	NA	Platform	0.6	Loomis & Craft 2010
GA	31°26'7.72"N	81°20'30.55"W	137 Cs	2001	1964-2001	NA	Platform	1.4	Loomis & Craft 2010
GA	31°27'7.52"N	81°21'56.94"W	137 Cs	2001	1964-2001	NA	Platform	0.3	Loomis & Craft 2010
GA	31°31'7.65"N	81°13'45.93"W	137 Cs	2001	1964-2001	NA	Platform	0.8	Loomis & Craft 2010
GA	31°32'9.39"N	81°17'44.17"W	137 Cs	2001	1964-2001	NA	Platform	4	Loomis & Craft 2010
GA	31°21'4.03"N	81°20'1.44"W	Feldspar	NA	2002-2018	NA	Platform	2.5	(C. Craft) Crotty et al., 2020
GA	31°23'14.77"N	81°16'50.35"W	Feldspar	NA	2002-2018	NA	Platform	3.2	(C. Craft) Crotty et al., 2020
GA	31°26'7.70"N	81°20'30.53"W	Feldspar	NA	2002-2018	NA	Platform	4.3	(C. Craft) Crotty et al., 2020
GA	31°27'7.53"N	81°21'56.94"W	Feldspar	NA	2002-2018	NA	Platform	1.8	(C. Craft) Crotty et al., 2020
GA	31°28'37.35"N	81°16'16.38"W	Feldspar	NA	2002-2018	NA	Platform	3.1	(C. Craft) Crotty et al., 2020
GA	31°31'7.65"N	81°13'45.91"W	Feldspar	NA	2002-2018	NA	Platform	3.7	(C. Craft) Crotty et al., 2020
GA	31°32'9.41"N	81°17'44.17"W	Feldspar	NA	2002-2018	NA	Platform	6.4	(C. Craft) Crotty et al., 2020
GA	31°21'4.03"N	81°20'1.44"W	SET	NA	2014-2018	NA	Platform	-1.5	(C. Craft) Crotty et al., 2020
GA	31°23'14.77"N	81°16'50.35"W	SET	NA	2002-2018	NA	Platform	0.2	(C. Craft) Crotty et al., 2020
GA	31°26'7.70"N	81°20'30.53"W	SET	NA	2002-2018	NA	Platform	0.3	(C. Craft) Crotty et al., 2020
GA	31°27'7.53"N	81°21'56.94"W	SET	NA	2002-2018	NA	Platform	0.1	(C. Craft) Crotty et al., 2020
GA	31°28'37.35"N	81°16'16.38"W	SET	NA	2002-2018	NA	Platform	0.5	(C. Craft) Crotty et al., 2020
GA	31°31'7.65"N	81°13'45.91"W	SET	NA	2002-2018	NA	Platform	-0.3	(C. Craft) Crotty et al., 2020
GA	31°32'9.41"N	81°17'44.17"W	SET	NA	2002-2018	NA	Platform	2.9	(C. Craft) Crotty et al., 2020
GA	31°57'58.61"N	81° 1'21.25"W	210 Pb	2011	NA	0.5	Platform	1.3	Alexander et al. 2017
GA	31°58'12.54"N	81° 1'11.57"W	210 Pb	2005	NA	0.65	Platform	2.7	Alexander et al. 2017
GA	31°58'16.07"N	81° 1'20.82"W	210 Pb	2005	NA	0.67	Platform	1.1	Alexander et al. 2017
GA	31°58'16.54"N	81° 1'20.54"W	210 Pb	2005	NA	0.6	Platform	1.7	Alexander et al. 2017
GA	31°58'18.70"N	81° 1'16.90"W	210 Pb	2011	NA	0.59	Platform	1.9	Alexander et al. 2017
GA	31°58'21.90"N	81° 1'32.95"W	210 Pb	2011	NA	0.67	Platform	1.8	Alexander et al. 2017

GA	31°58'22.51"N	81° 0'56.45"W	210 Pb	2011	NA	0.97	Platform	2.9	Alexander et al. 2017
GA	31°58'24.06"N	81° 1'13.12"W	210 Pb	2011	NA	0.87	Levee	3.4	Alexander et al. 2017
GA	31°58'27.48"N	81° 1'17.62"W	210 Pb	2011	NA	0.71	Levee/Platform	1.7	Alexander et al. 2017
GA	31°58'4.87"N	81° 0'55.48"W	210 Pb	2011	NA	0.69	Platform	3.7	Alexander et al. 2017
GA	31°58'5.41"N	81° 1'29.50"W	210 Pb	2011	NA	0.63	Platform	1.2	Alexander et al. 2017
GA	31°58'8.04"N	81° 1'45.48"W	210 Pb	2002	NA	0.23	Mudflat	1	Alexander et al. 2017
SC	32°30'19.61"N	NA	SET	NA	2008-2014	NA	Platform	2	Raposa et al. 2016
SC	32°20'43.10"N	80°27'56.80"W	SET	NA	2010-2017	0.878	Platform	2.01	(W. Doar) Crotty et al., 2020
SC	32°22'54.51"N	80°26'42.67"W	SET	NA	2010-2017	0.905	Platform	3.79	(W. Doar) Crotty et al., 2020
SC	32°29'41.15"N	80°19'26.41"W	SET	NA	2010-2017	1.036	Platform	4.33	(W. Doar) Crotty et al., 2020
SC	32°32'54.66"W	80°35'42.02"W	SET	NA	2010-2017	0.835	Platform	9.62	(W. Doar) Crotty et al., 2020
SC	32°32'54.94"N	80°35'44.70"W	SET	NA	2010-2017	0.835	Platform	-2.04	(W. Doar) Crotty et al., 2020
SC	33°19'56.18"N	79°10'18.07"W	137Cs	1982	NA	NA	Platform	2.5	Sharma et al. 1987
SC	33°19'56.59"N	79°11'5.35"W	137Cs	1982	NA	NA	Platform	1.3	Sharma et al. 1987
SC	33°18'50.78"N	79°11'37.50"W	210 Pb	1993	NA	NA	Platform	3.5	Vogel et al. 1996
SC	33°19'22.53"N	79°11'47.56"W	210 Pb	1993	NA	NA	Platform	1.6	Vogel et al. 1996
SC	33°19'36.89"N	79°12'6.41"W	210 Pb	1982	NA	NA	Levee	4.5	Sharma et al. 1987
SC	33°19'56.18"N	79°10'18.07"W	210 Pb	1982	NA	NA	Platform	2.4	Sharma et al. 1987
SC	33°19'56.59"N	79°11'5.35"W	210 Pb	1982	NA	NA	Platform	1.6	Sharma et al. 1987
SC	33°20'9.80"N	79°10'25.39"W	210 Pb	1993	NA	NA	Platform	2.9	Vogel et al. 1996
SC	33°21'0.39"N	79°11'31.66"W	210 Pb	1982	NA	NA	Platform	1.4	Sharma et al. 1987
SC	33°20'45.73"N	79°11'44.01"W	210 Pb, 137 Cs	NA	NA	NA	Platform	2.7	Raposa et al. 2016
SC	NA	NA	Feldspar	NA	11/1997- 1/1998	NA	Platform	2.97	Morris et al. 2002
SC	33°20'45.07"N	79°11'43.68"W	SET	NA	2008-2014	NA	Platform	2.6	Raposa et al. 2016
SC	32°51'55.67"N	79°42'06.10"W	SET	NA	2010-2017	0.705	Platform	-0.32	(W. Doar) Crotty et al., 2020
SC	32°52'52.96"N	79°40'27.96"W	SET	NA	2010-2017	0.6945	Platform	1.87	(W. Doar) Crotty et al., 2020
SC	32°57'53.03"N	79°37'57.03"W	SET	NA	2010-2017	0.797	Platform	3	(W. Doar) Crotty et al., 2020
SC	33°04'34.50"N	79°26'13.00"W	SET	NA	2010-2017	0.415	Platform	-3.97	(W. Doar) Crotty et al., 2020
SC	NA	NA	SET	NA	1997-2002	NA	Platform	5.1	Morris et al. 2002

Comment 1-10: L176-178: This seems like an odd place to introduce a hypothesis. It would be better in the introduction and used to motivate field experiment 1.

This section was removed from the main text and is now in the Supplement. We have removed the hypothesis from text and instead incorporated hypotheses into Figure 1, presented at the end of the Introduction. Thank you for the suggestion.

Comment 1-11: L179: Was this done on a spring tide?

This experiment was deployed on a spring tide. We now include tidal height in the manuscript.

Comment 1-12: L185-193: Is this intended to suggest that these redistribution rates persist across additional tidal cycles? Have other marsh studies in this system found significant flow-associated transport on the marsh platform? This runs counter to many studies indicating limited redistribution on vegetated marsh platforms. Or is the redistribution facilitated by other fauna such as crabs?

Given that our field experiments were only run for one tidal cycle to ensure we could reliably trace the fluorescently dyed material (i.e. over more tidal cycles the material can become so dispersed that it is no longer traceable), we do not know how the redistribution rates measured in this experiment may vary over additional tidal cycles. We now better summarize these constraints on the inference we can draw from this experiment in the text.

Lines 198-201: Due to challenges in confidently detecting the spatial extent of the fluorescently-tagged material beyond the single-tide duration of these experiments, additional work is needed to resolve how mussel biodeposits may be redistributed across the marsh over longer-time scale.

Moreover, the reviewer is right that our experimental results do not align with some prior work that emphasizes limited sediment redistribution in salt marshes, work that did not focus on areas influenced by mussel aggregations (see Comment 1-2 for citations and further context). However, we are confident that our methods are reliable in capturing both the surficial mass transport and redistribution of feces and pseudofeces locally deposited by mussels and the influence of mussels in facilitating sediment deposition in areas off of mounds via the transport and eventual settlement of their feces/pseudofeces.

Likewise, benthic crabs in particular have the potential to both directly (by trapping sediment on their leg hairs and redistributing this material) and indirectly (via excavating sediment and enhancing its potential for redistribution) facilitate mass sediment transport across the marsh surface as well, although that was certainly not the focus of this study. We now better describe how our results complement the literature by stating:

Lines 380-386: For example, endobenthic invertebrates, such as lugworms (*Arenicola marina*) and burrowing ghost shrimps (*Callinassa sp.*), commonly act as bioturbators, altering sediment cohesion, stability, and resuspension, as well as nutrient chemistry through their continued processing and displacement of sediment (Figure 7C; 36, 79, 80). Through these activities, bioturbators may play a key role in broadly conveying sediment captured by filter and suspension-feeders—although the magnitude and direction of these interactions are likely context dependent.

Comment 1-13: L210: Was tidal submergence 2x longer at the creekhead site?

While submergence is longer, we do not have measurements of the exact submergence times in each of these zones. However, we have previously measured submergence times at two tidal creekheads that comprised a range of elevations. For this revision, we compared across the two creekhead locations, and found that the lower elevation tidal creekhead (+0.605 mASL) was submerged for 3% longer than the higher elevation tidal creekhead (+0.655 mASL). In this *very* limited case, an increase of 0.05m in elevation was equivalent to a 3% increase in submergence time. Extending this rough estimate to comparisons across the two marsh zones used (which differ by an estimated 0.10-0.20m), we suggest submergence time may instead vary by 9-12% across the two marsh zones under consideration. We note that this is a very rough estimation, and does not include fluid dynamics known to control the temporal patterns in water and sediment transport across marsh platforms in the region. Ultimately, the difference in sediment deposition across these two zones is likely the result of both increased submergence times and higher concentration of food in the water column.

Comment 1-14: L238, Fig. 3: Is the width of the model (50 m) representative of the spacing of tidal creeks? If not, then the “entire marsh domain” deposition and accretion is representative of the tidal creek landscape but not the full marsh landscape.

The answer to this question depends on the marsh and tidal creek geomorphology. To address this question in the context of our model, we now include data on tidal creekhead density per 1km² of marsh habitat across the region. Across 10 sites in the region, we calculate a mean value of approximately 110 creekheads per square kilometer (see Comment 1-1 above). Assuming a creekhead area of influence of 50m x 50m, then we can calculate the proportion area of marsh platforms that tidal creekheads are likely to occupy:

$$\frac{\text{Area of one creekhead (2,500m}^2\text{) x Number of creekheads per km}^2\text{ (110 creekheads)}}{\text{Marsh platform total area (1km}^2\text{ or 1,000,000m}^2\text{)}}$$

We find that creekheads occupy approximately 27.5% of marsh area. Within our existing model, the creekhead area was 2,500m², while the whole domain (excluding the 1,000m² of main channel) was 9,350m². Therefore, in our initial simulations the creekhead was 26.7%. Therefore, we suggest that this is representative of both the tidal creek landscape as well as the full marsh landscape.

We acknowledge that a sensitivity analysis is useful to verify that we are not missing key interactions that may result from the inclusion of multiple creeks. We therefore ran additional model iterations in which we considered a domain consisting of three tidal creeks and their creeksheds (see Supplemental Methods). Each creek and creekshed has the geometry reported in Figure 5A, so the domain size is 150 m and 207 m in the long-shore and landward directions, respectively. We chose a density of mussel aggregation equal to 10% of the creekhead area, and the boundary conditions applied to the model were those described in the section “Methods - Delft3D Model”. The max difference between deposition and accretion we computed for one creek in the 3-creek watershed (i.e., within one-third of the total 3-creek watershed) and the ones we computed from a single creek creekshed (and reported in the paper) is 2.5%. Given the small differences in results, and the already large scope of this work, we suggest that these results support our choice of using one single creek and creekshed throughout the paper.

Comment 1-15: L242: How long were the simulations? What tidal conditions? I see the Methods answer these questions but it is difficult to understand the results without knowing more about the simulations.

We apologize for the lack of clarity and have revisited the Results section and included additional methodological information wherever feasible/necessary.

To answer your question, the simulations are 16 days long. This duration was chosen using the method proposed in Gray et al. (2021) which is fully described in the Methods section. This method defines the average spring + neap period in a year, which we assume as representative of the average hydrodynamic conditions in the study area, and consequently, of the mean morphological evolution of the salt marsh. The yearly morphological evolution of the salt marsh is then calculated by multiplying the results obtained from the 16-day simulations by the number of spring+neap cycles present in a year. This step is necessary since the computational effort needed for a yearlong simulation at the resolution required for our questions is not feasible. The tide is imposed at the channel boundary and corresponds to the astronomical tide during the average spring+neap period, reconstructed using all the tidal constituents calculated at the closest NOAA station (Daymark #156, Head of Mud River, GA" # 8674975).

To clarify this, we added the following part to the manuscript (Line 261-263):

Lines 261-263: All model simulations encompassed 16-day long periods, representing a yearly average Spring+Neap cycle (65). Annual values are calculated as the product of the results obtained from the 16-day simulations and the number of cycles present in a year.

65. Gray, M.W., Pinton, D., Canestrelli, A., Dix, N., Marcum, P., Kimbro, D., Grizzle, R., 2021. Beyond Residence Time: Quantifying Factors that Drive the Spatially Explicit Filtration Services of an Abundant Native Oyster Population. *Estuaries and Coasts*. <https://doi.org/10.1007/s12237-021-01017-x>

Comment 1-16: L249-251: These results suggest the effects of mussel-related enhanced deposition are very local and that sediment is not broadly redistributed through the system. I learned on L291-292 that the model "assumes no conveyance of biodeposits across marsh landscapes" [not noted in the Methods that I could see]. Why? If redistribution is central to the point of the paper, why not include redistribution in the model? It seems important in understanding the potential of hydrodynamic factors to disperse the biodeposits.

We agree and thank the reviewer for this thoughtful comment. We hope that we have satisfactorily addressed this concern in our response to Comment 1-2.

Comment 1-17: L254: Is -0.01cm/y the accretion rate for the creekhead and -0.08cm/y the accretion rate for landscape or is this a range of values for both related to scenario? If the former, why would deposition vary between sites by a factor of 2 and accretion by a factor of 8?

We have clarified the text and the figure to more clearly report the results (Lines 280-291). We apologize for the original error/lack of clarity.

Regarding the different magnitudes, the accretion rates are the average values obtained in the creekhead area and the entire domain, while the sediment deposition values are summed across these areas. From a mathematical point of view, the ~50% difference in sediment deposition between the two areas becomes

a $\sim 1/8$ reduction in accretion rate because the area of the domain is 4-times larger than the creekhead area.

From a morphodynamic point of view, the results are telling us that, even if we consider an area that is 4 times bigger than the creekhead, the sediment deposition simply doubles. This happens because most of the sediments that enter the marsh system deposit in the creekhead area. This happens for two reasons: (i) because the water (and then the sediments) preferentially enter the marsh system in that area; and (ii) because a large portion of these sediments are retained in that area by the mussels, due to filtration.

Comment 1-18: L258 and Fig. 3D,E: I am having trouble matching up values in text with values in the figure.

Thank you so much for pointing this out. We have fixed the figure and clarified the results. We apologize for this oversight.

Comment 1-19: L260-262: Nice result.

Thanks!

Comment 1-20: L280: “2020 vegetation-filtered DEMs” is a little confusing, without having read the Methods. Something like “DEMs developed from drone imagery in 2020, filtered to remove vegetation, ...”

We have now edited the wording to reflect these changes.

Comment 1-21: L284: How do these results compare with values from filters?

See response to Comment 1-9 for calculations.

Deposition on filter papers suggested accretion rates ranging from $+0.01 \text{ cm yr}^{-1}$ to $+0.74 \text{ cm yr}^{-1}$. Results from Experiment 2 ranged from 0.015 cm yr^{-1} to 0.487 cm yr^{-1} . See Comment 1-9 above for details. In comparison, the relative rates of accretion in Experiment 3 suggested that the mussel removal creekshed lost elevation at a rate of -1.7 cm yr^{-1} , while the mussel addition creekhead gained elevation at a rate of $+0.4 \text{ cm yr}^{-1}$. Therefore, these results are similar in magnitude and align with the predictions from our earlier experiments.

Comment 1-22: L287-289: Do mussels affect the geotechnical properties of the sediment or the presence of other fauna that could affect erosion rates when they are removed?

We find this comment very interesting! Yes, it is possible that the removal of the mussels affects both the geotechnical properties of the sediment as well as the presence and abundance of associated microbes and bioturbating fauna. Specifically, when mussels are present, both feces and pseudofeces are bound in mucus as they are deposited on the marsh surface—which certainly alters the geotechnical properties, such as erodibility. In addition, mussel aggregations are hot spots of secondary production, and facilitate a suite of soil microbes and benthic infauna, such as fiddler and Sesamid crabs. The loss of facilitation provided by mussel aggregations also likely decreases the amount of resuspension that occurs when these bioturbating organisms occur at high densities.

Disentangling these effects are unfortunately out of the scope of this work, but we do include additional text highlighting these specific complexities of how mussel removal may indirectly affect the fate of sediment. We allude to these direct and indirect influences on sediment geotechnical properties in lines 371-374.

Comment 1-23: L292: Does the model in any way capture the rugosity of the mussel mounds and the potential for enhanced deposition simply by trapping sediment and preventing resuspension? Were any experiments with nonliving mussel mound structures carried out to see how much the structure itself contributes to enhanced deposition?

We address this comment in two parts; first, we address the inclusion of rugosity in the model. Second, we discuss experiments that shed light on the effects of nonliving mound structures.

1. The influence of mussel mounds on sediment deposition is captured in the model through their influence on trapping sediments (i.e., via their filter feeding and biodeposition). In addition, the model considered the additional rugosity associated with the presence of taller vegetation above the mussel mounds. Taller and denser vegetation generates lower values of Chézy (i.e., higher rugosity), which is translated into lower flow velocities above the mussel mounds. We did not however also integrate the effect of smaller, partly-buried structures created by mussel shells on sediment deposition in the model—which we interpret to be what the reviewer is interested in here. Implementing additional rugosity of the mounds (i.e., hundreds of millimeter or centimeter variations across a 1m² area) would require a far greater degree of micro-topography and was out of scope of what was feasible for us to implement in the model for this study.

2a. In terms of experimentation, we note that for Experiment 3 we left behind all mounded pseudofeces and cordgrass (taller and denser than on the adjacent platform) when we removed mussels (see Lines 605-607). Thus, while we did not add in non-living shell structures, the majority of the physical mound structure and surficial rugosity was left intact.

2b. Further, in our previous work, we have added non-living shells as procedural controls in a mussel addition experiment and tracked effects on primary production (see Bertness et al 2015). This work found no effect of mussel shells on primary production, and while not reported in this 2015 paper, we only observed increased sediment deposition in plots with live mussels added (and not in those plots where only shells were added). In this current study, we did not repeat a similar experiment to assess the structural effect of the shells only in trapping sediment and thus cannot at this time disentangle the relative influence of the active biodeposition of material provided via mussel filter feeding versus the passive deposition of material associated with just their shells/structure. Given that mussels nearly continuously filter and biodeposit material when submerged and we can visibly observe the process of mussels building mounds of feces/pseudofeces within only hours after transplanting live mussels into a location without mussels, we anticipate that we have captured the most important processes (e.g. their filtration of material and biodeposition) in the model and in our experiments.

Bertness M, Brisson C, and Crotty S (2015). Indirect human impacts turn off reciprocal feedbacks and decrease ecosystem resilience. *Oecologia*, **178**, 231-237.

Comment 1-24: L301: Needs rewording: “of particularly of inorganic material”

We have changed the wording in this sentence. It now reads as follows:

Lines 335-336: We demonstrate that these faunal engineers substantially contribute to salt marsh sediment budgets, enhancing deposition by up to an order of magnitude.

Comment 1-25: L302: The results do not convince me that the enhanced deposition is “redistributed across marsh platforms”, which implies to me something more than local redistribution.

We agree and now specify in this statement that we are referring to ‘local’ deposition and not deposition more broadly.

Comment 1-26: L304-305: This statement also seems too strong. There is nothing in the results about SLR, which may affect the system in a multitude of ways that could impact the role of biodeposition. E.g., can localized high deposition create drainage issues that contribute to pond formation? The hydrodynamic aspects of this seem like something the model could address.

We agree with the reviewer. Our study does not explicitly analyze the interaction between SLR and mussel aggregations—and the consequences for marsh geomorphic evolution. Nonetheless, we suggest that the presence of mussel aggregations do not directly create drainage issues that lead to pond formation. The main reason is that ponds usually develop far from the creeks and channels (Mariotti, 2016). These areas are not populated by mussel aggregations, which preferentially grow close to the creek where submergence time and food availability is higher. Moreover, in these creekhead locations, mussel mounds are randomly staggered and do not develop a barrier to water flow. Such a barrier could prevent water to reach some areas of the marsh, and lead to pond formation.

However, we agree with the reviewer that further research is needed to assess whether longer-distance, indirect interactions between mussel aggregations and interior marsh areas can influence pond formation and maintenance processes. Given that interior ponds are not a very common feature in SE US Atlantic salt marshes, we do not specifically highlight this process in the text. However, we do clarify the influence of mussels on geomorphic processes by adding the following sentence to the manuscript:

Lines 342-345: However, further research is needed to address the long-term and large-scale interaction between mussel aggregations, marsh accretion, and sea level—and what the consequences of such interactions are for salt marsh morphodynamics and stability in the face of sea-level rise.

G. Mariotti, Revisiting salt marsh resilience to sea level rise: Are ponds responsible for permanent land loss? *J. Geophys. Res. Earth Surf.*, **121**, 1391–1407 (2016).

Comment 1-27: L313, Fig. 5: It is not clear why fauna other than mussels are being brought into the picture here. This would be more appropriate for a review-type article on engineering fauna.

We appreciate the comment here and we include more results-focused text in our discussion. However, we disagree that any mention of other systems is inappropriate. We feel that, since many fauna are predictably patterned and have disproportionate effects on the flow of sediment in systems, our results do have implications for how researchers focused on other systems may consider faunal engineering effects on geomorphic evolution. For a journal with a broad readership like *Nature Communications*, it is common for studies to summarize how their results may extend to other systems as we do.

Comment 1-28: L320: Should this sentence about indirect effects follow the text on L314-317 about indirect effects?

We have reorganized the text to accommodate this recommendation.

Comment 1-29: L324: To be convinced that the contribution of mussels to sediment deposition over space and time can be predicted, I'd have to be convinced that the distribution and density of mussels can be predicted. The paper doesn't address that. In fact, it is noted as necessary next step in L345.

We apologize for the lack of clarity and we think this is a critical point of our paper. We now more explicitly explain the hierarchy of factors controlling mussel population presence and density (see Comment 1-1 for full details).

Comment 1-30: L351: "coastal ecosystems"

This change has been made.

Comment 1-31: L363: It is hard to visualize the way the filters are deployed. How is the wire used to attach the filter to the petri dish? How is the petri dish affixed to the marsh or mussel aggregations. Is the petri dish level? How high is the lip on the petri dish? Perhaps a photo could be added to the Supplementary figures to show how this was done.

We have added photos and a complete description below.

Material:

- Polystyrene Petri Dishes (Falcon 351029) 100 x 15 mm: Fisher Scientific, Catalog No. 08-757-100D
- Whatman Quantitative Filter Paper, Grade 42 Circles, ashless grade: Fisher Scientific
- Steel wire, \approx 1.5 mm gauge, that can be cut into \approx 25 cm long pieces and bend at 90° angles

Step 1: Drill two small holes in petri dish.

Step 2: Cut and bend wire, so that it will hold petri dish and filter paper in place.

Step 3: Label each filter paper. Place filter paper over petri dish and poke holes with wire where the filter covers the holes in the petri dish.

Step 4: Remove filter paper from petri dish. Weigh filter paper and write down filter paper number and weight. Place filter paper on top of petri dish and secure with wire.

Step 5: Cut 24 X 15 cm aluminum sheets and wash with acetone.

Step 6: Fold into a square, fold the sides and label. Weigh the pouch and record the weight.

Step 7: Deploy the filter on a flat, undisturbed location. The filter should be flush with the marsh surface, with the lip of the petri dish inserted into the sediment. Each filter has an assigned pouch. When the filter is removed from the marsh it is stored in its assigned pouch (without the petri dish).

Step 8: Back in the lab, open the pouch, but leave the filter inside and place both in an oven (< 50° C) to dry. Place in a desiccator to cool down before weighing both filter and pouch together.

Comment 1-32: L373: These filters were deployed without petri dishes?

These filters were deployed with petri dishes. This has been noted in the method section.

Comment 1-33: L381: Needs rewording: “in persistent in marsh”

This wording has been changed for clarity.

Comment 1-34: L426: Regarding the marsh blocks being “washed completely clear of all surface sediment”, is there no surface sediment on the marsh platform? Can this be done without disturbing sediment below the surface? It is hard to imagine what the marsh blocks looked like after cleaning.

On the surface of the marsh, there is a layer of unconsolidated sediment (often including pseudofeces or feces). Below this material is a thick mat of consolidated peat, i.e., belowground roots and rhizomes. Of course, within this mat there is bound sediment which we do not disturb. Instead, we used power-washing with water and finer scale WaterPik machines to remove all unconsolidated surficial sediment, such that the water draining from the washing process runs clear. See schematic below for a detailed description of the process.

Comment 1-35: L459: How sensitive are the results to the density of mussels in a mound? Do all mussel mounds found on creekbanks have a similar number of mussels? Do they all filter a similar amount? Does it vary by life stage, temperature, other factors?

Mussel aggregations found at creekheads occur in a range of sizes (i.e., number of mussels they include). Large mounds with up to several hundred individuals typically occur closer to the creekhead, and they decrease in size with distance onto the marsh platform (see Comment 1-1 above). To approximate these dynamics and to enable us to address our primary research question, we quantified an average creekhead mussel density representative of the entire 2,500m² creekhead area. To calculate this value, we surveyed 8 mussel aggregations at each of 3 distances from tidal creekheads (0m, 20m onto marsh platform, and 40m onto marsh platform) at ten creeks distributed across sites on Sapelo Island and the surrounding marshlands (n= 240 total mussel aggregations). We recorded total number of mussels in

each aggregation as well as the mound dimensions. We then calculated the average mussel density per square meter and found this value to be 177 mussels m⁻². However, there is a great deal of variation, such that the standard deviation is 88 mussels.

To assess how sensitive the model results are to mussel density, we ran a new set of model iterations. We selected a scenario where mussels occupy 20% of the marsh area (to amplify their effects and potential differences across mussel density treatments). We then established three mussel density treatments, 89 mussels m⁻² (mean – 1SD), 177 mussels m⁻² (mean), and 265 mussels m⁻² (mean + 1SD) and compared the sediment volume deposited in the whole domain in each scenario. We find that the effects of mussel density are not negligible and vary almost linearly with sediment volume contribution (and resulting accretion rate). Compared to the no-mussel scenario, the low mussel density scenario increases total sediment deposition in the domain by 1.33%, as compared with the intermediate density at 2.00%, and the high density at 2.55%. However, given that the mean value is representative of the average mound size, and that the relationship between mound size and contribution is linear, we are comfortable using this value to provide an accurate estimate of their role in the greater creekhead area.

We acknowledge that there is certainly more complexity that could be included in future versions of the model. For example, there is evidence that filtration rate can vary with body size (*S. Williams, unpublished data*), temperature (Moody and Kreeger 2019), parasitic load (*J. Morton, manuscript in prep*), and other factors. However, as inclusion of this complexity would require additional data to parametrize the model, and it is not central to our main research question addressed in this paper, we suggest that it is out of the scope of this work at this time.

J Moody, D Kreeger, Ribbed mussel (*Geukensia demissa*) filtration services are driven by seasonal temperature and site-specific seston variability. *Journal of Experimental Marine Biology and Ecology*, **522**, 151237 (2020).

Morton J.P., Davis B.P., Walker T., Haber I., Adelson E. Silliman B.R. *In prep*. Parasitism disrupts a keystone facultative mutualism in a coastal ecosystem. Target Journal: *Ecological Monographs*. Anticipated submission: December 2022

Comment 1-36: L464: I don't think tidal creek SSC is ever noted in the paper, so the error is hard to evaluate. It would be helpful if tidal creek SSC at different tidal stages was provided. How sensitive are the results to the value of settling velocity?

The settling velocity was obtained by fitting the suspended sediment concentration (SSC) measured at different instants of the tidal cycle, by using the equation $C_s = C_{s0} e^{-t \cdot w_s / h}$, where w_s is the settling velocity [m s⁻¹], h is the slow depth [m], C_{s0} is the initial suspended sediment concentration [kg m⁻³], and t is the time [s] (see figure below).

We modified the sentences in Lines 525-531 as follows: "This value provides the best fit of the Total Suspended Sediment (TSS) concentration we surveyed in a creek, on the adjacent Sapelo Island, with an error of 0.022±0.025 kg m⁻³ (MAE+RMSE). The fit was obtained by using the exponential decay formulation that reads:

$$C_s = C_{s0} e^{-t \cdot w_s / h}, \quad (1)$$

where w_s is the settling velocity in [m s^{-1}], h is the slow depth in [m], C_{s0} is the initial sediment concentration in [kg m^{-3}], and t is the time in [s]. We set C_{s0} equal to 0.10 g m^{-3} , which approximates the average value measured during flood tide, at the same location and tidal cycle. In addition, we set h equal to 0.30 m , which is the local mean annual high tide, calculated for 2018.”

We also added the following figure and caption in the Supplement:

Figure S8. Exponential fitting of the Total Suspended Sediment (TSS) concentration data surveyed at Dean Creek (blue dots), located in Sapelo Island, Georgia, USA. The time is in seconds from the reference time, which corresponds to the time of the first survey performed at slack water.

As for computing the sensitivity of the results, we have run a simulation in which we increased settling velocity by 50% (i.e., to settling velocity equal to 0.15 mm s^{-1}), and extra deposition due to mussel mounds varied by only approximately 6.5% of the original value.

Comment 1-37: L474: Is the overall area covered by mussels held constant in the sensitivity analysis or is it varied? Are the results sensitive to the total area of mussels?

Yes, the area covered by mussel mounds has been kept constant while varying the numerical cell size. We performed the sensitivity analysis separately for each of the three configurations (i.e., mussel mounds covering 0, 10, and 20% of the creek head). Our analysis indicates that results are not sensitive to grid size for each of the percent coverage scenarios (see Lines 542-543).

Comment 1-38: L540: This must have entailed many trips across the marsh. How was disturbance to the marsh surface minimized?

We discuss the effects on the marsh and the efforts to minimize them in the Reporting summary. Our responses are pasted below for ease of consideration:

Disturbance: Minimizing disturbance was key to the success of our experimental deployments, as movement of sediment was what we aimed to capture. Thus, in all instances, the number of field personnel was limited to those necessary to complete work. All physical steps were carefully executed such that cordgrass was not trampled and damaging trails were not established and maintained. For all movement of heavy blocks of sediment or large number of mussels, sleds were utilized at high tide (or on rising tides) to aid in movement and to minimize disturbance to the site. For excavation of sediment to allow for insertion of sediment catchment devices, all shoveling/excavation was carefully conducted from within the plot boundary such that no edges were trampled. For the large-scale mussel manipulation, it was not possible to completely exclude the use of trails because of the large extent of the manipulation. However, we limited the establishment of trails to areas outside the creekhead area (2500m^2). These

trails are visible in aerial imagery in 2018 but are no longer present by 2020, suggesting that the disturbance was minimal.

Comment 1-39: L558: “did inform”?

We rephrased this sentence.

Comment 1-40: L826: Isn't this the same as Fig. 5A?

This is the prevailing literature, notably excluding the effects of fauna. Figure 5A includes the effects of fauna on each of these important ecogeomorphic factors.

Comment 1-41: Random question: Do mussels inhibit or accelerate headward creek extension?

We agree that this is a very interesting question and a topic we have discussed at multiple instances over the last few years. Due to the reasons we summarize below, we were unable to address this potential pathway by which mussels may influence the geomorphic evolution of these marshes in this paper. First, to address this question, a large scale, well-replicated and long-term (e.g. 5+ year long) field experiment akin to that which we set up at one site and evaluate for a few years in Experiment 3 is needed. This is because our prior work evaluating creek elongation reveals that there is significant heterogeneity from creek to creek in their extension rates due to creek features (e.g. their length, width, depth) as well as surrounding landscape features (e.g. does the creek fill and drain a large or small marsh area). Given this heterogeneity one would need to manipulate mussels at the creekshed scale across multiple creeks and run the experiment for long enough that treatment effects could be detected. Given that it took our team several months to set the one site for experiment 3, an experiment of this scale and duration was not logistically feasible. Moreover, without field data to parameterize the model, we felt that any model simulations of the influence of mussel effects on creek elongation could not be well-verified with field data. Thus, there are excellent opportunities to explore such dynamics in future work but we felt that we could only evaluate the influence of mussels on sediment deposition well in the current study.

Reviewer #2 (Remarks to the Author):

I have reviewed the manuscript entitled 'Faunal engineering stimulates landscape-scale accretion in southeastern US salt marshes' (NCOMMS-22-26048). This manuscript describes a series of five aligned experiments and modelling efforts which together assess the scale and density dependent role of mussel aggregations in marsh sediment deposition and accretion. The manuscript represents a novel combination of field data and spatial models and I particularly applaud the authors for their investigation that spans multiple spatial scales which are critical to inferring real-world effects of organisms on coastal ecosystem function and geomorphology. The manuscript presents substantial evidence for the role of mussel aggregations in promoting marsh accretion both at local site and estuary wide scales and a variety of indirect and direct effects of mussels are attributed to this role.

The results are an important addition to the literature and interesting. However, I have a few suggestions to improve readability, understanding, and interest:

Comment 2-1: Sea-level rise linkage: I find the linkage between the pace of sea-level rise and rates of accretion are interesting and an important implication of this work. However, the linkages as they are

stated in the manuscript are tenuous without some further exploration of the assumptions of sea-level rise changes to marshes – for example, sea-level rise is unlikely to solely change water depth, it is likely to change hydrodynamic regimes and therefore there will probably be interactions with rates of accretion in marshes. Perhaps in the discussion, the authors could add a little more nuance to the linkage between their results and the implications for changes under various sea-level rise projections.

Absolutely, we agree. Please see Comment 1-26 above relating to sea level rise and pond formation.

We have also included a more nuanced discussion of potential influence of sea level rise on mussel-mediated sediment deposition and associated marsh accretion. For ease, we have copy and pasted the relevant paragraph below:

Lines 336-345: Our implementation of the Delft-3D BIVALVE module and subsequent simulations further emphasize that mussels can successfully be incorporated into models of coastal wetland accretion, and together with the large-scale mussel manipulation, we highlight that their landscape effects are demonstrably contributing to marsh vertical growth. Although not explicitly evaluated in this study, we project that mussels, by bolstering sediment deposition and marsh accretion, likely play a key role in modulating the future persistence of these intertidal ecosystems with accelerating sea-level rise. However, further research is needed to address the long-term and large-scale interaction between mussel aggregations, marsh accretion, and sea level –and what the consequences of such interactions are for salt marsh morphodynamics and stability in the face of sea-level rise.

Comment 2-1: Results and clarification: It is a challenge and an art to present results in the stand alone format required for this journal and compounding that, this paper includes methods and results for five different studies that could all form papers on their own. I see a lot of merit in presenting the findings from all five together as it builds clear evidence for the profound effects of mussels on coastal geomorphology and function at multiple scales and contexts. However, as is, the results are difficult to follow because a lot of the methodological detail is buried either in the supplement or the methods at the end. These issues may be able to be resolved with some edits to clarify (see below for detailed comments). But arguably, the manuscript perhaps tries to fit too much detail for each experiment/survey/model, and it could be best to focus on one key finding from each study that contributes evidence to the overarching multi-scale story. The authors might also consider one overarching conceptual figure to tie all five studies together, which could be a map that has the scale of each experiment/model marked and the key findings summarized for each. This would also emphasize the novelty of this approach.

We thank the reviewer for this comment and have built a new conceptual figure that outlines the experimental and/or field components, summarizing their locations, methodological key points, temporal and spatial scale, and driving hypothesis (see Comment 1-3 and new Figure 1). We additionally simplified the results to focus on a key finding (or a few) key findings rather than many (e.g., move discussion of inorganic and organic sediment to the supplement), and added methodological information throughout to improve readability.

Examples where clarification is required:

Comment 2-2: - The terminology to describe the sampling positions is confusing to follow in the text (e.g., outer-marsh levee, levee, levee crest, levee-adjacent, non-mussel high marsh platform, aggregation-adjacent, etc.), and it also appears that the terminology is not consistent throughout the manuscript (e.g.,

compare position descriptions on lines 142-151 with 354-361). This may seem like a minor issue, but it makes the results difficult to follow and interpret.

This terminology is now streamlined throughout. We have additionally included several new conceptual figures to help the reader follow the studies that are included in the the main manuscript and supplement.

Comment 2-3: Figure 1 – I am not sure how to interpret this figure – for example, the caption states that the bars with black borders are the mussel aggregation data, but what are the bars without black borders around them? The scale bar has two colors, but what do those colors represent because the bars in the figure are only one color? Perhaps also label all the position descriptors that are used in the text here? - In places, there is not enough methodological information in the results to interpret the figures, and some important details are buried in the supplementary material; for example, Figure 2C shows a dotted line for the ‘open control’ results, but it is only when I look at the supplementary Figure 3 that I can see what an ‘open control’ is. Further, the open controls have no mussels, but the line spans the mussel gradient on the x axis which is confusing.

We thank the reviewer for highlighting these issues with the figures and we have made every attempt to clarify the results, improve figure legends, and better highlight key results. Please see the updated figure and figure legend below.

Figure 3. Filter paper results. (A) Summer neap, (B) summer spring, (C) winter neap, and (D) winter spring tide results are presented across 13 marsh location types. Mean sediment deposition is presented as gray bars (mean \pm SE) on each marsh location, with letters denoting statistically significant differences among treatments (Tukey HSD, $p < 0.001$). Results collected from atop mussel aggregations are presented as gray bars with a black border (at 0m, 10m, and 20m from the tidal creekhead). Results from locations not associated with mussel aggregations (i.e., all other marsh location types) are presented as gray bars with no border.

Comment 2-4: Experiment 2 – the written text on line 203-217 refers to the different positions as ‘low’ and ‘high’ tidal elevations, but in Figure 2 and in other places in the text they are referred to as ‘Creekhead’ and ‘Platform’. This is another example of inconsistency in terminology which makes the results difficult to follow.

We apologize for the inconsistent terminology. We have ensured that all terminology now remains consistent throughout the manuscript and have adopted the Creekhead and Platform naming convention.

Comment 2-5: In Figure 2C – which relationships do the linear lines correspond to? The relationships with or without cordgrass?

As there was no effect of cordgrass, the linear lines are inclusive of all plots of 6 treatments and represent the relationship between mussel biomass on the x axis and sediment deposition on the y axis. The horizontal line represents the last treatment, the no-mussel, no-cordgrass control (0M, 0C). We have now clarified this in the legend.

Comment 2-6: Line 280 – Acronym DEMs is not defined earlier

DEM is now defined earlier in the text (Lines 313-314).

Comment 2-7: Line 200-202 and 417-421 – Instead of ‘intermediate size mussel aggregations plus no cordgrass’, could you simplify treatment names to a series of “+mussels/-cordgrass” treatment labels? E.g., ++++mussels/-cordgrass (80 mussels and 0 cordgrass)? This naming system would improve ability to follow the different treatment effects in the written text.

We have changed the naming conventions (Lines 208-213 and 472-477).

Comment 2-8: Line 440 – A 1 cm mesh cloth was attached to the sediment trap boxes, but the manuscript doesn’t state why this mesh is required.

The mesh cloth was included to allow invertebrate access to and from the mound. We did this because during prototyping of the devices, several small fiddler crabs fell into the vertically oriented PVC poles and could not climb out. If the device was left for multiple days, these crabs died of physical stress at low tide. Therefore the mesh was required to ensure that all weight that was accumulated was due to sediment and not fauna.

Reviewer #3 (Remarks to the Author):

This is an exciting and creative study with surprising results, highlighting the importance of mussels on marsh accretion. However, I do have a number concerns before I can recommend its publication.

Comment 3-1: One particular concern is the missing vegetation effect on sediment deposition in both your experimental work and the modeling. If I did not misunderstand, you even demonstrate negative effects of vegetation presence on sedimentation. This clearly contradicts with the available experimental and modeling literature. How do you explain this? Can you be certain that your results on mussel effects are robust when your experiments and your modeling cannot replicate these well-established biogeomorphic feedbacks?

Thank you for this comment, and yes, this is something we discussed as a team throughout the process.

As we state on lines 269-274, vegetation causes a reduction in marsh sediment deposition, in comparison to a scenario where the marsh has no vegetation. The reviewer correctly states that the most emphasized bio-geomorphic feedback of vegetation consists of increased friction, reduced velocity, and increased deposition.

However, many other studies demonstrate that vegetation does not always increase sediment deposition in a salt marsh or coastal environment. This happens because vegetation increases surface roughness, and consequently can divert flow and sediments toward less vegetated areas or reduce the exchange of water and sediments between high and low elevation areas of the system. Since in this study we did not consider sediment resuspension, the increase in surface roughness is the only variable driving sediment transport and deposition.

In the revised manuscript, we now state (Lines 269-274): While the most emphasized bio-geomorphic feedback of vegetation consists of increased friction, reduced velocity, and increased deposition (66-68), this negative effect of vegetation on deposition has been observed both numerically and experimentally (34; 69-72). Mechanistically, the negative effect of vegetation occurs when the aboveground biomass diverts flow and sediments toward less vegetated areas or reduces the exchange of water and sediments between high and low elevation areas of the system.

Comment 3-2: Along the same lines, I miss data on the vegetation density (or structure, or standing biomass). I did not find any data on this. How do they relate to field data or those presented in other mesocosm experiments (i.e. marsh organs)?

This comment refers to the cordgrass biomass present (or removed) from the internal unit of our sediment catchment devices. We have addressed this question, summarized the data, and included new text and a new figure to address this concern. Please see Comment 3-10 for a complete description of the steps we took.

I attach a pdf with some additional comments.

Comment 3-3: page 4, lines 94-99: Ren et al. 2022. Ecol Letters. provide a good overview from a blue carbon perspective

We now include this citation in Line 95.

Comment 3-4: page 4, lines 100-103: Mueller et al. 2019 MEPS demonstrate how herbivores affect the spatial distribution of allochthonous vs. autochthonous organic inputs

We now include this citation in Line 102.

Comment 3-5: page 6, line 140: Is this a standard procedure? How does this measure relate to actual rates of sediment deposition or accretion? Has it been calibrated against other methods? Please provide more detail and reference to other studies.

Using filter paper to measure sediment deposition is a standard procedure in the literature, dating back to the late 1980s (Reed 1989). We provide two of the seminal papers on this method below. Please refer to comments 1-9 and 1-31 above regarding more detail on the methods employed as well as how these measures related to sediment accretion measured by different methods across the region.

Reed, D. J. 1989. Patterns of sediment deposition in subsiding coastal salt marshes, Terrebonne Bay, Louisiana - the role of winter storms. *Estuaries* 12:222-227.

Reed, D., T. Spencer, A. Murray, J. French, and L. Leonard. 1999. Marsh surface sediment deposition and the role of tidal creeks: Implications for created and managed coastal marshes. *Journal of Coastal Conservation* 5:81-90.

Comment 3-6: page 7, line 158: For minerogenic marshes (e.g. NW Europe) season is the key driver with almost all sediment being deposited during fall/winter. Please elaborate here (or in the discussion) if you see your results to be generalizable across organogenic (peat-forming) and minerogenic marsh systems.

The reviewer brings up an important point related to whether our results have relevance to both minerogenic and organogenic marshes. Certainly, the specific mechanism by which mussels are actively filtering suspended organic and inorganic material and depositing it onto the marsh is unique to our southeastern US salt marsh focal system. In this region, total suspended solid concentrations are high relative to other regions and mussels are fairly well distributed across the marsh landscape, allowing for their faunal engineering effects to be quite important (e.g. they can deposit large masses of material over relatively large areas). In other systems in which TSS values are lower and faunal engineers have more constrained spatial distributions, their influence on sediment deposition and accretion would likely be muted.

On the topic of minerogenic versus organogenic, we would argue that faunal engineers have the potential to be important to both marsh types. In the case of minerogenic marshes, faunal engineers can influence sediment deposition and retention while in organogenic marshes they have the potential to influence (in potentially positive or negative ways), the accumulation of organic material such as through their ability to stimulate or inhibit belowground root production. We now make a more intentional statement about these influences on both marsh types in lines 371-378 (pasted below).

Lines 371-378: One tractable pathway for such expansion is to further employ the Delft-3D model and adapt the BIVALVES module we develop in this study to probe questions about the future geomorphic evolution of these marshes and other vegetated coastal ecosystems where faunal engineers influence plant, sediment and hydrodynamic processes. In particular, there may be particular value in exploring the relative importance of bivalves and other faunal engineers in modifying the accretion processes and rates of organogenic coastal wetlands, where land formation and growth strongly primarily depends on plant

growth (particularly of roots and rhizomes) and secondarily upon sediment deposition, versus minerogenic wetlands that are more strongly dependent on sediment deposition.

Comment 3-7: page 8, lines 182-183: Does the choice of material matter here? Above you state that particularly inorganic material is deposited by mussels. Here you use organic matter.

Pseudofeces and feces (biodeposits) are comprised of both organic and inorganic material. We do not distinguish between these two fractions in the redistribution experiment.

Comment 3-8: page 8, lines 190-193: How about more effective sediment trapping as an alternative/additional mechanism explaining the mussel effect?

Please see our response to comment 1-23 above.

Comment 3-9: page 8, line 197: A technical drawing or better photo would be useful to understand this thing better.

We now include a more technical drawing of the sediment catchment device in Figure S4 (pasted below). Photographs of the devices deployed are also presented as part of Figure S5.

Sediment Catchment Device

Side View

Central box: 36 cm x 36 cm x 16 cm

Outer box: 61 cm x 61 cm x 8 cm

- Packed with 1-in. and 2-in. diameter PVC (3-in. length) to catch all sediment deposited in box

Bird's Eye View

Figure S4. Schematic of sediment catchment device. For experiment 2, sediment blocks were cut to 36cm x 36cm x 16cm and placed within plastic-encased bins of the same dimensions. Bins containing marsh blocks were then centrally placed and fitted within an additional larger bin (61cm x 61cm x 8cm), with the top of each box flush to the same height (see Side View, left). The outside bin was filled with 64, 5-cm diameter PVC poles and 32, 2.5-cm diameter PVC poles (both 8-cm in height) so that all bin edges were held upright and PVC was rigidly filling all space within the outer box (see Bird's Eye View, right). These sediment catchment units were then transported back to the experimental site where recipient holes were dug to the exact dimensions, so that the top of the marsh block (along with the top of each PVC pole) was exactly flush with the marsh surface sediment.

Comment 3-10: page 8, line 201: How much cordgrass and how does it relate to field densities or other mesocosm (ie marsh organ) studies? This information is not sufficient to assess the (relative) importance of mussels vs. vegetation.

All aboveground cordgrass biomass from the 36 x 36cm marsh block was cut at the marsh surface, cleaned, dried, and weighed at the end of the experiment. Aboveground cordgrass biomass was higher in all replicates that had mussel aggregations, both from the creekhead (89.2 ± 11.3 g / plot) and on the marsh platform (91.7 ± 15.5 g / plot). In plots without mussels, the aboveground cordgrass biomass was 54.8 ± 8.0 g / plot on the creekhead and 42.9 ± 7.0 g / plot on the marsh platform. We suggest that this is representative of cordgrass biomass in the field in these zones (see Crotty & Angelini 2020 for additional data points; Figure S6, pasted below).

Marsh Organ Literature Comparison

Marsh organs in the literature vary in size. For example, the marsh organs employed by Kirwan and Guntenspergen (2015; link) were 15cm diameter PVC pipes, and the aboveground biomass contained in the pipes ranged from 0g to <40g for both *Schoenoplectus americanus* and *Spartina patens*.

In contrast, Mega-Marsh Organs (MMO), such as those deployed by Cao et al. 2021 have internal dimensions of 75cm x 45cm (length*width) x 30cm in depth. Within the MMOs, the dry biomass of *Spartina anglica* ranged from <5g to ~25g (treatment means; see Figure 5 in Cao et al. 2021, link).

For *Spartina alterniflora*, the species used in our paper, Andrew Payne and colleagues (2021) used 10-cm diameter PVC pipes as their marsh organ device and found a range of ~5-8 grams of aboveground cordgrass biomass per plot. When we scale up the 5g found in these 10cm diameter marsh organs (78.5cm² surface area) to the area we used in our experiment (1,296cm²), we find a value of 82.5 g. This is exactly in line with the data we found, which ranged from 42.9 to 91.7 grams per plot (surface area 1,296cm²).

Comment 3-11: page 9, lines 211-213: This result contradicts with most of the available literature. How was this result handled with (also in your modeling)?

Please see our reply to Comment 3-1 (above). We do want to clarify that we do not intend to say that there is no effect of vegetation on deposition. We acknowledge that removing vegetation over small patches surrounded by larger vegetation areas does not capture the landscape effect of vegetation on

sediment deposition, since suspended sediment particles would likely be captured by the surrounding vegetation, whose effects in reducing turbulent kinetic energy over marsh surfaces are well known (e.g., Christiansen et al., 2000; Mudd et al., 2010). To capture the effects of vegetation absence at the landscape, experimental removal of vegetation would need to occur over much larger areas (e.g., Temmerman et al., 2012)—which is what we aimed to explore with our Delft-3D model. Future research should focus on analyzing the impact of vegetation removal on marsh deposition, by using coupled field and numerical approaches. We now clarify this in the text.

Lines 228-229: To accurately capture the effects of vegetation presence beyond the patch scale, however, experimental removal of vegetation would need to occur over much larger areas (e.g., 62).

Mudd, S.M.; D'Alpaos, A.; Morris, J.T. How does vegetation affect sedimentation on tidal marshes? Investigating particle capture and hydrodynamic controls on biologically mediated sedimentation. *J. Geophys. Res. Earth Surf.* 2010, 115, 1–14, doi:10.1029/2009JF001566.

Christiansen, T.; Wiberg, P.L.; Milligan, T.G. Flow and sediment transport on a tidal salt marsh surface. *Estuar. Coast. Shelf Sci.* 2000, 50, 315–331, doi:10.1006/ecss.2000.0548.

Temmerman, S.; Moonen, P.; Schoelynck, J.; Govers, G.; Bouma, T.J. Impact of vegetation die-off on spatial flow patterns over a tidal marsh. *Geophys. Res. Lett.* 2012, 39, doi:10.1029/2011GL050502.

Comment 3-12: page 9, line 221: Is this model applicable to both organogenic East coast marshes and minerogenic NW European marshes? Does sediment load play a role?

The model has the potential to be adapted to both organogenic as well as minerogenic marshes, although for application to the former processes related to root biomass production/decomposition and other forms of organic matter cycling would need to be included. Of note, the marshes in the Southeastern US are more minerogenic (i.e. their average organic content of the top 10cm of soil is typically below 15%) likely due to very high decomposition rates as well as high suspended sediment concentrations. To avoid confusing our readers, we did not add any text specifying this constraint to the model in the text.

Comment 3-13: page 9, line 231: why only two levels? What is the density/standing biomass stock?

The two levels that were considered for the Delft3D model were 1) no vegetation, and 2) vegetation representative of the standing stock measured from experimental sites on Sapelo Island. The vegetation stem density, stem height, and stem diameter varied by marsh area type. The characteristics of each area type are reported in Table S2. We utilized only these two levels because they were the key scenarios relevant to our driving question: What is the relative role of cordgrass (and mussel) biomass in driving sediment deposition? No other cordgrass scenarios were required to address the topics contained within this manuscript; we do agree that another paper should address how the changing vegetation structure (i.e., extension of low marsh into higher elevations) with sea-level rise will affect projected contribution to sediment deposition across marsh zones.

Comment 3-14: page 10, lines 245-246: This contradicts with or current understanding of vegetation effects. How do you explain this? I do not see this discussed below.

See above reply to Comment 3-11.

Comment 3-15: page 12, lines 306-307: contradicting your results, right?

See above rely to Comment 3-11.

Comment 3-16: page 17, line 422: How important are mineral contributions in this system?

Please see comments 3-6 and 3-12 above. As previously noted, our focal marsh system – similar to much of the coastline from the Florida/Georgia border to North Carolina are fed by relatively large rivers that contribute fairly high sediment loads to the coast. Thus, the mineral contributions are significant. To better highlight this aspect of our study region, we emphasize the sediment-rich rivers that feeds the Georgia coast in the main text (see Lines 144-147).

Comment 3-17: page 17, line 423: Specify, otherwise it is hard to understand the missing vegetation effects.

We have addressed this in the text (Lines 479-482).

Reviewer comments, second round -

Reviewer #1 (Remarks to the Author):

The authors have done a commendable job of responding to my comments as well as those of the other reviewers. The addition of the new figures 1 and 2 help clarify the regional context, the study area, and the experimental design and methods. Removing or moving to the supplement the sections on regression tree analysis and organic content significantly helped to streamline the results section and improve readability. The points I thought had been overstated have been revised and clarified. Overall, I think this is a very interesting set of studies and will make a nice contribution to the literature on the impacts of fauna on saltmarsh depositional rates and patterns.

A few minor points:

L215: Now that Fig. 2 shows the SE coastal region, can the location of this study be more specifically identified rather than just "in a Georgia salt marsh"?

Fig. 3: Can the legends in the lower right of the shaded areas be moved to someplace in the upper left where the background is white? It is difficult to read the green text on the gray background.

L294: I may have lost a thread in the paper, but it wasn't clear to me what aspect of resuspension increasing spread through the marsh had been predicted ["resuspension increases sediment spread through the marsh as predicted"], nor what "spread through the marsh" was intended to suggest about whether the spread was local or broad. It didn't appear that any more information about these sensitivity runs was included in the supplement with the other sensitivity runs, but perhaps something could be added there.

L377: "strongly primarily depends" is an awkward construction.

L413, L431: It isn't necessary to sequence the steps using "To next ..."; remove the "next"

L519: concentration in the water column?

Reviewer #2 (Remarks to the Author):

The revised manuscript is much improved and in particular the new Figure 1 is a welcome addition that integrates and ties together the five studies across spatial scales.

Outstanding comments to consider:

Figure 4C and caption - the dotted line spans the mussel gradient on the x axis, but this line represents a zero mussel treatment. I think this line should only cross the x axis at 0 mussel biomass? There are data points for the + cordgrass treatment and these appear to span the mussel biomass gradient - how can this be so, when the +cordgrass treatment was only placed at 0 and 50M mussel treatments and not the whole gradient?

Line 498-499 - Arguably, having no mesh would allow invertebrates access. I think the addition of mesh is to limit the entry of large animals that can affect the amount of sediment measured in the traps? The authors might consider re-wording.

Thank you for the opportunity to review an interesting and novel manuscript.

Reviewer #3 (Remarks to the Author):

The authors thoroughly addressed all my comments.

Now that I understand how the Catchment Devices work, I have another thought concerning the

missing vegetation effect on sediment deposition. Sediment deposition was quantified in a ring of PVC pipes surrounding the plant/mussel/control square in the center of the device. Did you also try to quantify sedimentation or accretion in the central square? Isn't it likely that plants caused more sedimentation directly where they grow, i.e. between their shoots? If so, what would be the implications, also with respect to the mechanism by which mussels increase sedimentation (filtering from water column vs. "simple" trapping)?

I would appreciate the authors reflect on this last point. Other than that, I recommend publication of the manuscript.

REVIEWERS' COMMENTS

Reviewer #1 (Remarks to the Author):

The authors have done a commendable job of responding to my comments as well as those of the other reviewers. The addition of the new figures 1 and 2 help clarify the regional context, the study area, and the experimental design and methods. Removing or moving to the supplement the sections on regression tree analysis and organic content significantly helped to streamline the results section and improve readability. The points I thought had been overstated have been revised and clarified. Overall, I think this is a very interesting set of studies and will make a nice contribution to the literature on the impacts of fauna on saltmarsh depositional rates and patterns.

We thank the reviewer for the thoughtful edits and comments throughout this process. We feel that it has substantially improved the paper.

A few minor points:

L215: Now that Fig. 2 shows the SE coastal region, can the location of this study be more specifically identified rather than just “in a Georgia salt marsh”?

Yes. We have made the suggested change.

Fig. 3: Can the legends in the lower right of the shaded areas be moved to someplace in the upper left where the background is white? It is difficult to read the green text on the gray background.

Absolutely. We have added a white box behind the legend and increased the font size to improve readability. Thank you for the suggestion.

L294: I may have lost a thread in the paper, but it wasn't clear to me what aspect of resuspension increasing spread through the marsh had been predicted [“resuspension increases sediment spread through the marsh as predicted”], nor what “spread through the marsh” was intended to suggest about whether the spread was local or broad. It didn't appear that any more information about these sensitivity runs was included in the supplement with the other sensitivity runs, but perhaps something could be added there.

We are not able to comment on whether the spread is likely to be local versus broad at this time. What the sensitivity analysis shows is that when you allow for resuspension on mounds (e.g., from bioturbation by infauna), then the sediment is redistributed to other areas of the marsh (i.e., the total deposition on the marsh platform is higher). We note that <1% of the resuspended sediment exits the marsh. We now clarify the wording of this section (Lines 297-298).

L377: “strongly primarily depends” is an awkward construction.

We have clarified the construction of this sentence.

L413, L431: It isn't necessary to sequence the steps using "To next ..."; remove the "next"

We have made these changes, thanks!

L519: concentration in the water column?

We have changed "on" to "in the water column."

Reviewer #2 (Remarks to the Author):

The revised manuscript is much improved and in particular the new Figure 1 is a welcome addition that integrates and ties together the five studies across spatial scales.

We thank the reviewer for their help and comments throughout this process.

Outstanding comments to consider:

Figure 4C and caption - the dotted line spans the mussel gradient on the x axis, but this line represents a zero mussel treatment. I think this line should only cross the x axis at 0 mussel biomass? There are data points for the + cordgrass treatment and these appear to span the mussel biomass gradient – how can this be so, when the +cordgrass treatment was only placed at 0 and 50M mussel treatments and not the whole gradient?

We have incorporated the suggested changes to Figure 4.

These points do span from approximately 6kg/m² to 14kg/m². The overall set of mussel treatments range from <1 to >20kg/m². The reason for the large variation in the "50 mussel" treatment is that all mounds are natural aggregations harvested from the field. Some mounds have a larger number of smaller individuals, while others are predominated by larger individuals. We elected to allow for some of this natural variation because we wanted to limit transplantation effects—and the resultant effect on the rates of biodeposition.

Line 498-499 – Arguably, having no mesh would allow invertebrates access. I think the addition of mesh is to limit the entry of large animals that can affect the amount of sediment measured in the traps? The authors might consider re-wording.

We have added this interpretation to the text (Lines 538-539).

Thank you for the opportunity to review an interesting and novel manuscript.

Thank you!

Reviewer #3 (Remarks to the Author):

The authors thoroughly addressed all my comments.

Now that I understand how the Catchment Devices work, I have another thought concerning the missing vegetation effect on sediment deposition. Sediment deposition was quantified in a ring of PVC pipes surrounding the plant/mussel/control square in the center of the device. Did you

also try to quantify sedimentation or accretion in the central square? Isn't it likely that plants caused more sedimentation directly where they grow, i.e. between their shoots? If so, what would be the implications, also with respect to the mechanism by which mussels increase sedimentation (filtering from water column vs. "simple" trapping)?

We did quantify the sediment that accumulated both in the surrounding bin of PVC tubes as well as in the center bin. We include this data below. In all panels, blue lines and circles reflect the data collected from the creekhead zone, while yellow lines and circles represent data from the marsh platform. Sediment deposited in the middle bin, or "center," is represented with a blue or yellow center with black or gray outline, respectively. Trendlines are presented as solid lines. In contrast, data collected in the outer bin, or "edge," is represented by a black or gray center with a blue or yellow outline. Trendlines are presented as dashed lines.

In panel a, we show dry mussel biomass on the x axis and dry sediment biomass on the y axis. Four trendlines are shown: 1) creekhead, data from "center" (blue, solid); 2) creekhead, data from "edge" (blue, dashed); 3) platform, data from "center" (yellow, solid); and 4) platform, data from "edge" (yellow, dashed). We see that when mussel biomass is 0, the volume of sediment deposited in the edge is much higher than the sediment deposited in the center (edge intercepts are 690 and 605 while center intercepts are 93 and 205). However, as you increase mussel biomass, the rate of increase in sediment deposition is ~2-times higher in the center than on the edge (slope in the center is 0.74 and 0.35, while edge is 0.37 and 0.19). We note that deposition in the edge and center is higher in the creekhead than on the platform.

In panel b, we show the relationship between dry cordgrass biomass on the x axis and dry sediment biomass on the y axis. All R² values are <0.08. Contrary to the reviewers point, the presence of cordgrass had a larger effect on the edge than in the center in both zones. In 2 of the 4 cases, the slope was negative (creekhead center and platform edge).

To remove the noise of the many plots without cordgrass, we also look at the relationship between cordgrass biomass and sediment deposition in just treatments with cordgrass, both with and without mussels. Again, the presence of cordgrass had a larger effect on edge

deposition, rather than deposition in the central bin. In the absence of mussels, 2 of the 4 treatments again had a negative slope (i.e., an increase in cordgrass correlated to a small decrease in sediment deposited). In the presence of mussels, an increase in cordgrass biomass was always correlated with an increase in sediment deposited—but with a large degree of variation. Given these results, we suggest that trapping of sediment by cordgrass plays a very small role in controlling sediment deposition at the small spatial scale that we were able to investigate.

I would appreciate the authors reflect on this last point. Other than that, I recommend publication of the manuscript.

We thank you very much for all of your help with this manuscript!